# Description and validation of the Japanese algorithm for radiative flux and heating rate products with all four EarthCARE instruments : Pre-launch test with A-Train

Akira Yamauchi[1], Kentaroh Suzuki[1], Eiji Oikawa[2], Miho Sekiguchi[3], Takashi M. Nagao[1], Haruma Ishida[4]

[1]Atmosphere and Ocean Research Institute, University of Tokyo, Kashiwa, Japan
[2]Meteorological Research Institute, Japan Meteorological Agency, Tsukuba, Japan
[3]Faculty of Marine Technology, Tokyo University of Marine Science and Technology, Tokyo, Japan
[4]Meteorological Satellite Center, Japan Meteorological Agency, Tokyo, Japan

*Correspondence to*: Akira Yamauchi (yamauchi.akira@jaxa.jp)

**Abstract.** This study developed the Level 2 (L2) atmospheric radiation flux and heating rate product by a Japanese team for Earth Cloud, Aerosol, and Radiation Explorer (EarthCARE). This product offers vertical profiles of downward and upward longwave (LW) and shortwave (SW) radiative fluxes and their atmospheric heating rates. This paper describes the algorithm developed for generating products, including the atmospheric radiative transfer model and input datasets, and its validation against measurement data of radiative fluxes. In the testing phase before the EarthCARE launch, we utilized A-Train data that provided input and output variables analogous to EarthCARE, so that the developed algorithm could be directly applied to EarthCARE after its launch. The results include comparisons of radiative fluxes between radiative transfer simulations and satellite/ground-based observations that quantify errors in computed radiative fluxes at the top of the atmosphere against Clouds and Earth's Radiant Energy System (CERES) observations and their dependence on cloud type with varying thermodynamic phases. For SW fluxes, the bias was 24.4 $Wm^{-2}$, and the root mean square error (RMSE) was 36.3 $Wm^{-2}$ relative to the CERES observations at spatial and temporal scales of 5 °and 1 month, respectively. On the other hand, LW exhibits a bias of -10.7 $Wm^{-2}$ and an RMSE of 14.2 $Wm^{-2}$. When considering different cloud phases, the SW water cloud exhibited a bias of -11.7 $Wm^{-2}$ and an RMSE of 46.2 $Wm^{-2}$, while the LW showed a bias of 0.8 $Wm^{-2}$ and an RMSE of 6.0 Wm-2. When ice clouds were included, the SW bias ranged from 58.7 to 81.5 $Wm^{-2}$ and the RMSE from 72.8 to 91.6 $Wm^{-2}$ depending on the ice-containing cloud types, while the corresponding LW bias ranged from -8.8 to -28.4 $Wm^{-2}$ and the RMSE from 25.9 to 31.8 $Wm^{-2}$, indicating that the primary source of error was ice-containing clouds. The comparisons were further extended to various spatiotemporal scales to investigate the scale dependency of the flux errors. The SW component of this product exhibited an RMSE of approximately 30 $Wm^{-2}$ at spatial and temporal scales of 40 °and 40 days, respectively, whereas the LW component did not show a significant decrease in RMSE with increasing spatiotemporal scale. Radiative transfer simulations were also compared with ground-based observations of the surface downward SW and LW radiative fluxes at selected locations. The results show that the bias and RMSE for SW are -17.6 $Wm^{-2}$ and 172.0 $Wm^{-2}$, respectively, which are larger than those for LW that are -5.6 $Wm^{-2}$ and 19.0 $Wm^{-2}$, respectively.

## 1. Introduction

Clouds and aerosols play a significant role in shaping the Earth's radiation budget, exerting a substantial influence on global climate. Changes in the planet's radiation balance can affect atmospheric circulation patterns (Hartmann and Short, 1980; Liebmann and Hartmann, 1982). In particular, the energy imbalances caused by aerosols and clouds, quantified by their radiative forcing, can affect atmospheric circulations and the transport of water vapor. However, their quantitative estimates are still subject to significant uncertainty. The commencement of observations by the Earth Radiation Budget (ERB) onboard NIMBUS-7 in 1978 marked an improvement in our understanding of the radiation budget at the top of the atmosphere (TOA) (Kyle et al., 1985). With the launch of the Earth Radiation Budget Experiment (ERBE) in 1984, observations of the radiation budget at the TOA became more precise (Barkstrom, 1984). Broadband Radiometer (BBR) observations at the TOA have continued since then and now provide valuable long-term data through instruments such as the Clouds and the Earth's Radiant Energy System (CERES, Wielicki et al.1996). Observations conducted by the BBR at the TOA were utilized to ascertain the Earth's overall energy budget. The radiation budget estimate is important not only at the TOA but also at the atmosphere (ATM) and surface (SFC) to quantify the energy partitioning into ATM and SFC, with implications for climate effects due to their differing heat capacities. Although radiation fluxes at the TOA can be directly observed by the BBR, observing fluxes at the ATM and SFC is more challenging. Typically, fluxes at the ATM and SFC are estimated using radiative transfer (RT) computations utilizing aerosol and cloud parameters obtained from various satellites as inputs (e.g. Rossow and Lacis, 1990; Rossow and Zhang, 1995; Zhang et al, 1995; Whitlock et al, 1995).  Space-based RT calculations are commonly used to estimate radiative fluxes within the atmosphere and at the surface, complementing surface-based measurements where direct observations may be limited or unavailable on the global scale.

Information about clouds and aerosols obtained from passive sensors has been utilized in RT computations to estimate the Earth's energy budget (e.g., Fasullo and Trenberth, 2008a, b). However, passive sensors do not provide vertical profile information, resulting in significant uncertainty in the estimation of the cloud base height, which particularly influences the longwave (LW) radiative flux. CloudSat (Stephens et al. 2008) and the Cloud–Aerosol lidar and Infrared Pathfinder Satellite Observations (CALIPSO; Winker et al. 2010) satellites were introduced to the A-Train satellite constellation and provided crucial vertical profile information about clouds and aerosols. The downward longwave radiation at the surface estimated by using the cloud vertical distributions observed by CloudSat and CALIPSO was found to be 10 $Wm^{-2}$ larger than model-based estimations in the decade of 2000s (Stephens et al., 2012). Recent studies have indicated that Earth's radiation budget is affected by the vertical distribution of clouds (Li et al., 2015; Oreopoulos et al., 2017), emphasizing the importance of understanding the vertical distribution of clouds. This improved understanding was achieved with atmospheric radiation products generated from the CloudSat/CALIPSO cloud and aerosol vertical information, such as 2B-FLXHR and 2B-FLXHR-Lidar (L'Ecuyer et al., 2008; Henderson et al., 2013) developed by NASA's CloudSat team, and Clouds and CERES-CALIPSO-CloudSat- Moderate-Resolution Imaging Spectroradiometer (MODIS) (CCCM, Kato et al., 2011) developed by the CERES team. The accuracy of these products based on active sensors was verified through comparisons with ground- and satellite-

based observations of radiative fluxes at the TOA and surface (L'Ecuyer et al., 2008). In addition, spaceborne active sensors have made significant contributions to climate science by providing more precise constraints on atmospheric and surface radiative fluxes compared to passive sensors. These active sensors play a crucial role in improving climate models by offering more accurate measurements of radiative fluxes and heating rates partitioned into atmosphere and surface (Haynes et al., 2013;

L'Ecuyer et al., 2019; Hang et al., 2019). However, without quantifying the uncertainties, it is difficult to fully evaluate the reliability of these estimates of radiation based on active sensors. Therefore, one of the objectives of this study is to assess these uncertainties through comparisons with other products and ground-based observations, aiming to validate the accuracy and reliability of the radiative flux based on the active sensor. The Earth Clouds, Aerosols, and Radiation Explorer (EarthCARE) satellite, scheduled for launch in 2024 as a "successor" to the CloudSat/CALIPSO satellites, will provide even

more detailed information on cloud and aerosol vertical distributions, particularly with Doppler capability of measuring vertical motion of cloud particles (Illingworth et al., 2015; Wehr et al. 2023). With the enhanced performance of EarthCARE's CPR and ATLID, which will better capture low-level clouds, we expect to see improved contributions to downward longwave radiation as well. These measurements will also provide important continuity for the long-term data record that began with the A-Train in 2006 (L'Ecuyer and Jiang, 2010), ensuring that trends and patterns in atmospheric observations are consistently

maintained. Novel EarthCARE information is expected to offer even more accurate pictures of Earth's energy budget in conjunction with cloud dynamics. Such enhanced information of Earth's energy budget will also be facilitated by improved knowledge of vertical profiles of radiative fluxes expected from the detailed cloud profiling capability combined with cloud dynamics information.

To this end, Japanese and European teams are independently developing aerosol, cloud, and radiation products for the

85 EarthCARE satellite mission, which will become available to the research community after its launch. This study describes an algorithm developed for generating one of the products offered by the Japanese team, namely the Level 2 (L2) atmospheric radiative flux and heating rate product listed in Kikuchi et al., (2019), including aerosol and cloud radiative forcings, and its validation against satellite- and ground-based measurements of radiative fluxes at the top of the atmosphere and at the surface. This product was generated using cloud and aerosol vertical profile information from the EarthCARE satellite instruments

of Cloud Profiling Radar (CPR), ATmospheric Lidar (ATLID), and multi-spectral imager (MSI), and was validated against measurements of radiative fluxes from the Broadband Radiometer (BBR) also onboard EarthCARE (Illingworth et al., 2015; Wehr et al., 2023; Eisinger et al., 2023; Okamoto et al., 2024). The L2 atmospheric radiative flux and heating rate product provides vertical profiles of the downward and upward radiative fluxes for LW and SW and their respective atmospheric heating rates. When employed alongside other Japanese L2 cloud and aerosol products, it is possible to discern the impact of

various cloud and aerosol characteristics, particularly in their vertical dimensions, on atmospheric radiation. A European team is currently developing an equivalent atmospheric radiation product (Cole et al., 2023). Once the data are released, cross comparisons with our dataset will be conducted to evaluate the products and enhance their accuracy. These products serve as fundamental data for a variety of climate research applications, including the evaluation and improvement of global climate models. The objective of this study was to develop and validate the accuracy of the Japanese team's L2 atmospheric radiation

algorithm and characterize the source of radiative flux errors when applied to real observed satellite measurement data. One particular criteria to be used for evaluating the EarthCARE science goals is whether the error between RT flux and observations remains within ±10 Wm$^{-2}$ (ESA, 2001). The uncertainty of ±10 Wm$^{-2}$ is associated with a spatial scale averaged over 100 km² and is based on instantaneous measurements. To provide fundamental information for such an assessment and broader scientific data use, it is important to investigate the extent to which the flux error varies with spatiotemporal scales. To this end, this study quantified the flux errors at various spatiotemporal scales.

In the testing phase, before the launch of EarthCARE, we utilized A-Train-based aerosol and cloud vertical profile data obtained from the CloudSat/CALIPSO/MODIS satellites, which provided data analogous to those expected from EarthCARE, as input data. This served as a preliminary evaluation prior to the official launch of EarthCARE. Knowledge of the aerosol component is crucial for assessing the aerosol radiative effect. In this regard, it is worth noting that the components of aerosols characterized may vary between A-Train and EarthCARE products. For instance, the number of aerosol components in the post-EarthCARE launch product has increased from three (water-soluble (WS), dust (DS), and sea salt (SS)) to four, incorporating WS, light-absorbing (LA), DS, and SS (Nishizawa et al., 2007, 2008, 2011, 2019; Kudo et al. 2023). By adding LA in the EarthCARE product, the estimation of light-absorption for biomass burning and air pollution, which include LA, will be improved and aerosol radiative effect is expected to be more accurately evaluated during the EarthCARE mission.

The remainder of this paper is organized as follows. Section 2 describes the atmospheric radiative transfer model, input dataset, and validation data. In Section 3 demonstrates an example of comparing input values with output values and CERES. In Section 4, the radiative forcings of clouds and aerosols are estimated based on the validated radiative fluxes to demonstrate how the radiative effects of aerosols and clouds can be quantified using this product. In Section 5, we present validation results by comparing our atmospheric radiation products with those from NASA in terms of flux errors relative to the CERES measurements of radiative fluxes at the TOA. The comparison is shown in the form of the spatiotemporal scale dependency of the flux errors and error characteristics broken down into different cloud types with varying thermodynamic phases on a global scale. In Section 6, the radiative fluxes are validated at the ground surface and the errors between the fluxes of this product and the ground-based fluxes of the Baseline Surface Radiation Network (BSRN) are presented. Finally, Section 7 summarizes the conclusions of this study.

## 2. Data and Methods

### 2.1 Algorithm overview

The radiative flux of this product was obtained through vertical one-dimensional (1D) radiative transfer calculations utilizing information on the vertical profiles of aerosols and clouds along with meteorological data as input. The 1D RT simulation was chosen as a method primarily due to the requirement for data processing latency in EarthCARE Japanese standard product. In this regard, it is worth noting that our previous study developed a three-dimensional (3D) RT code and applied it to some cases

of cloudy scenes as described in Okata et al. (2017). It would then be possible in future studies to seamlessly compare the 1D and 3D RT calculations based on the common assumptions and settings of particulate and gaseous optical properties that are used in our 1D and 3D RT codes. This would allow for error quantifications of 1D against 3D RTs and possible introduction of several methods for approximating 3D effects in the framework of 1D RT computation, as also described in Okata et al. (2017). In future studies, we plan to incorporate these improvements into the standard algorithm with 1D RT described in the present paper, as well as to develop a radiative flux algorithm based on 3D RT calculations (Okata et al. 2017) as part of the research product, so as to add values to our Japanese radiation products. During the pre-launch test phase, the A-train satellite constellation was used as a substitute for EarthCARE. Aerosol vertical profile data from the CALIPSO satellite and cloud vertical profile data from CloudSat/CALIPSO/MODIS were used for this purpose. After the launch of EarthCARE, aerosol profile data obtained from ATLID and cloud profile data acquired from CPR/ATLID/MSI will be utilized. CloudSat/CALIPSO/MODIS and CPR/ATLID/MSI will provide consistent cloud and aerosol parameters. This product offers SW and LW radiative fluxes, their respective radiative heating rates, and the radiative forcing of clouds and aerosols. The accuracy of this product was validated by comparing radiative fluxes with those measured by available satellite instruments (CERES in A-Train and BBR in EarthCARE) at the TOA and with those obtained from BSRN at the surface. In the next section, the details of the input data and radiative transfer model used are described.

## 2.2 Input Data and radiative transfer model

In the analysis of this paper, we utilize data from the A-Train constellation at the time of writing this paper before the EarthCARE data becomes available. While EarthCARE products will be used for future operational applications, A-Train data, including observations from CloudSat, CALIPSO, and MODIS, are currently employed to evaluate and refine the algorithm in preparation for application to the EarthCARE data. The A-Train data provides a valuable proxy for the type of information that will be available from EarthCARE, although there are some differences in instrument characteristics and data resolution. These differences are taken into account in our analysis to ensure that the results are relevant for the upcoming EarthCARE mission. For the EarthCARE mission, the algorithm will utilize data from the CPR_CLP (from CPR), the ATL_CLA (from ATLID), and the MSI_CLP (from MSI). These instruments provide vertical profiles of clouds and aerosols, which are critical inputs for calculating radiative fluxes and heating rates. The A-Train data, on the other hand, allows us to test and validate the algorithm using observations that are similar in nature to those expected from EarthCARE, ensuring that the algorithm is robust and ready for operational use once EarthCARE data becomes available.

Input data for aerosols were obtained from the Japan Aerospace Exploration Agency (JAXA) LIDAR Aerosol Property Product (CA-Aprop) (Nishizawa et al., 2007, 2008, 2011). The extinction coefficient of fine-mode spherical particle (WS), coarse-mode spherical particle (SS), and non-spherical particle (DS) are derived from the CALIPSO observation. The vertical profiles of extinction coefficient at 532 nm for WS, DS, and SS are used in the radiative transfer calculations. The particle size and optical properties of these three aerosol components are listed in Nishizawa et al. (2011).

The input data for ice clouds were obtained from JAXA's Radar/Lidar Cloud Microphysics Property Product (CSCA-Micro) (Okamoto et al., 2010; Sato and Okamoto, 2011), which provides information on the ice particle effective radius and Ice Water Content (IWC). The input data for the water clouds, that is, the effective particle radius and Cloud Water Content (CWC) for liquid clouds, were obtained from NASA's CloudSat 2B-CWC-RO-R04 product (Austin et al., 2009). NASA's product was used in our current developmental version of the algorithm because of the lack of a long-term CWC dataset in the current version of the JAXA CSCA-Micro product. However, it will be replaced by JAXA's CWC in the operational product after EarthCARE is launched.

The total cloud optical thickness (COT) was also used to better constrain cloud radiative properties, particularly for SW. When there is a discrepancy between the COT derived from the vertical information of CloudSat/CALIPSO and the COT from MODIS, the vertical extinction coefficient is adjusted to align with the COT from MODIS. The COT data were obtained from JAXA's Moderate Resolution Imaging Spectrometer (MODIS) Imager Cloud Property Product (MOD-Micro, Nakajima and Nakajima 1995; Nakajima et al., 2019; Wang et al., 2023). Voronoi particles (Ishimoto et al., 2010) were assumed in the COT retrieval of ice clouds from MODIS. The Voronoi particles are particles that do not have regular spherical shapes, but rather particles with irregular polyhedral shapes. In cases where the MODIS COT was available, the extinction coefficient for each vertical layer originally derived from the CloudSat/CALIPSO was rescaled by the total COT derived from MODIS. When the COT was not available from MODIS, the extinction coefficient derived from the CloudSat/CALIPSO data was used directly. The meteorological field variables (pressure, temperature, specific humidity, and skin temperature) from NASA's Goddard Earth Observing System (GEOS-5) Data Assimilation System (Bloom et al., 2005; Rienecker et al., 2008) are used in the radiative transfer calculations. The ground surface albedo data used in this study were obtained from the MODIS global albedo product, MCD43C3 (Roesch et al., 2004; Schaaf et al., 2002). Sea surface albedo was set to 0.05. In this study, a constant sea surface albedo is used to simplify the radiative transfer calculations and to minimize computational demands. This assumption is based on the understanding that variations in sea surface albedo have a relatively minor effect on uncertainty of the overall radiative flux compared to other variables such as cloud cover and aerosol properties.

All input data, including aerosol and cloud profiles, meteorological field variables, and ground surface albedo, were averaged to a horizontal resolution of 5 km and 34 vertical layers prior to their use in the radiative transfer calculations. The original resolution of all the individual datasets, including cloud, aerosol, surface, and meteorological fields, is 1 km × 240 m. MODIS global albedo product (MCD43C3) is gridded at a 0.05° by 0.05° spatial resolution. This averaging is primarily due to the computational cost of radiative transfer for meeting the latency requirement of data processing and is also for consistency with the footprint of BBR and CERES, which is around 10km and 20km, respectively. In this averaging, if even a single grid within the 5 km grid contains clouds, the cloud profile for the entire 5 km grid is treated as uniformly cloudy, with values averaged horizontally. The original product is designed with a 1 km footprint resolution, but the 5 km grid assumes horizontal uniformity of cloud distribution within the grid, and values are averaged accordingly to account for any inhomogeneity. Radiative transfer simulations were performed using the one-dimensional radiative transfer (RT) model of MstrnX (Sekiguchi and Nakajima, 2008; Nakajima et al., 2000). As an assumption of the ice cloud optical properties, Voronoi particles were used to account for

the non-spherical shape of the ice particles in both the JAXA/ATrain product and the EarthCARE mission (Wang et al., 2023). This assumption of ice particles in the RT simulation was consistent with that of the MODIS and MSI ice cloud retrievals. The results for the radiative fluxes obtained from the radiative transfer calculations were compared with data from the CERES measurements (CER_ES8_Aqua-FM3_Edition3) at the TOA, the Baseline Surface Radiation Network (BSRN, Ohmura et al., 1998) measurements at the surface, and the CloudSat/CALIPSO 2B-FLXHR-Lidar products of versions R04 and R05 (L'Ecuyer et al., 2008; Henderson et al., 2013) at the TOA for validation. All comparisons, including those with other products and BSRN data, were conducted using instantaneous data. In 2B-FLXHR-Lidar R05, the input values for clouds and aerosols have been updated to use the R05 versions of the CloudSat products. These updates include improvements in cloud coverage, cloud physical properties, including updated cloud phase information, and the use of CALIPSO V4 products for aerosols, which update the global distribution of aerosol components and aerosol optical depth (AOD). These enhancements allow for more accurate flux calculations. In line with both this product and other NASA flux products, only the daytime flux data were utilized for the four-month comparison period of January, April, July, and October, 2007. The presence of aerosols and clouds can significantly influence radiation calculations and requires verification under various conditions. Therefore, this study compared all-sky conditions, including all clouds observed, and cloudy conditions of different types with varying cloud phases. When verifying, the all-sky conditions include instances where the 20 km area was a mixture of clear and cloudy conditions, while the cloudy conditions extract only instances where the entire 20 km area was covered by clouds. Under all-sky conditions, comparisons were made even when data were available only from CloudSat/CALIPSO (not from MODIS) to maximize cloud samples with vertical profiles. However, for cloud-type classification using the thermodynamic phase, only cases with available MODIS data were considered for comparison. For comparison with 2B-FLXHR-Lidar regarding different cloud types, the 2B-CLDCLASS-Lidar R05 (Sassen et al., 2008) and MOD06-1KM-AUX R05 (Platnick et al., 2017) products were used for cloud type classification. To ensure a consistent comparison between the RT simulations and CERES measurements, the simulated radiative fluxes were averaged over a 20 km area to match the CERES footprint. The CERES flux data is 20 km x 20 km including both along-track and cross-track directions; however, the 1D radiative transfer calculation compares the flux calculated only in the along-track direction, so the comparison with CERES requires consideration of this point. When comparing with BSRN, the surface measurement time closest to the CloudSat transit time within ±0.1 deg of the footprint was selected. The time was matched such that the time difference between the A-Train transit and ground observation was no more than 10 min at maximum.

## 2.3 Demonstration of input and output

To better understand the capabilities of the algorithm described in this section, we present an example of its application. Specifically, we demonstrate how the algorithm works to produce radiative fluxes and heating rates using major inputs of aerosols and clouds. Figure 1 illustrates the vertical profiles of aerosols and clouds that serve as critical inputs for the radiative transfer model. These profiles are essential in determining the accuracy of the calculated radiative fluxes and heating rates. By examining this example, we highlight the conditions that can lead to flux errors, providing a clearer understanding of the

algorithm's performance. Figure 1 shows a comparison of the radiative fluxes at the TOA between a radiative transfer calculation and the observed values. Figure 1 (a) shows the cloud top phase of MODIS observation along with the A-Train satellite orbit and Figure 1 (b) and 1 (c) show the vertical profiles of aerosols and clouds used as inputs of the RT model. The computed radiative flux was subsequently used to determine the radiative heating profile (Figure 1 (d)). The accuracy of the radiative fluxes and radiative heating rates was verified through a comparison with observations at the TOA. Figure 1 (e) and (f) compare the calculated and CERES-observed fluxes for LW and SW, respectively. The latitudinal resolution in panels (b) to (f) of Figure 1 is shown at 5 km. By examining these outputs, we can observe the impact of different atmospheric conditions on the radiative transfer calculations. The spatial distributions of aerosol and cloud radiative effects, as seen in later figures, highlight the significant variability across different regions. This understanding is crucial for interpreting the subsequent evaluation of the algorithm's accuracy. This step-by-step illustration of the algorithm provides a clear understanding of its operational process, setting the stage for a more detailed evaluation of the results. The LW radiation was generally consistent with the observations, regardless of the cloud or aerosol distribution. However, SW radiation shows a greater discrepancy from the observations, particularly when clouds are present. This greater discrepancy in the presence of clouds is likely attributable to the significant influence of cloud optical properties and their dependence on the thermodynamic phase used as the input information. In the subsequent sections, we present a comparison of our RT simulations and other products with CERES measurements on a global scale and their classification into cloud thermodynamic phases to investigate the flux errors of our product in more detail.

To quantify the flux errors of the RT simulations against observations, the bias and Root-Mean Square Error (RMSE) relative to observations were calculated as follows:

$$\text{Bias} = \frac{1}{n}\sum_{i=1}^{1}[(\text{FLUX}_{\text{RT}})_i - (\text{FLUX}_{\text{obs}})_i] \, , \tag{1}$$

$$\text{RMSE} = \left[\frac{1}{n}\sum_{i=1}^{1}[(\text{FLUX}_{\text{RT}})_i - (\text{FLUX}_{\text{obs}})_i]^2\right]^{1/2} , \tag{2}$$

where $\text{FLUX}_{\text{RT}}$ denotes the radiative flux calculated by the RT model, and $\text{FLUX}_{\text{obs}}$ denotes the observed flux value. The fluxes are then converted into radiative heating rate using the following equation:

$$\frac{dT}{dt} = \frac{g}{C_p}\frac{dF}{dp}, \tag{3}$$

where $T$ is the temperature (K), $t$ is time (s), $g$ is the acceleration due to gravity (m s$^{-2}$), $C_p$ is the specific heat content of air at constant pressure (J kg$^{-1}$ K$^{-1}$) with $C_p$ = 1005 J kg$^{-1}$ K$^{-1}$, $F$ is the radiative flux (W m$^{-2}$), and $p$ is the pressure (Pa).

## 3. Aerosol and Cloud radiative forcing

The L2 radiation product provides not only radiative flux and heating rates but also instantaneous radiative forcing due to aerosols and clouds. Aerosol radiative forcing (ARF) and cloud radiative forcing (CRF) are calculated as the difference between the radiative fluxes with and without aerosols or clouds, respectively. Specifically, ARF is defined as the difference between the radiative flux calculated with all aerosol components included and the flux calculated without aerosols. Similarly, CRF is defined as the difference between the radiative flux with all cloud components included and the flux calculated in the absence of clouds. These calculations are performed for both the TOA and the SFC to assess the impact of aerosols and clouds on the Earth's energy budget. The following figures (Figures 2 and 3) illustrate the spatial variability in the radiative effects of aerosols and clouds as calculated by the algorithm. These visualizations provide context for the algorithm's performance in representing the Earth's energy budget. The algorithm based on RT simulations with aerosols and cloud input, as described above, can be used to estimate aerosol and cloud radiative forcing, which are key factors in evaluating the impact of aerosols and clouds on the energy balance. Figures 2 and 3 show the global distributions of the aerosol and cloud radiative forcing, respectively, as examples of the output from the algorithm. The radiative forcing shown was derived from the accumulation of instantaneous flux values with and without aerosols or clouds. Radiative forcing can be classified into SW and LW components at different levels: TOA, within the atmosphere (ATM), and at the SFC. Net radiative forcing is obtained by combining the SW and LW radiative forcing components.

The results in Figure 2 reveal the distinct characteristics of aerosol radiative forcing in the TOA and SFC. In the atmosphere, both SW and LW radiative forcing demonstrated varying responses to the presence of specific aerosols, resulting in either positive or negative radiative forcing. This finding emphasizes the influence of aerosol optical properties on the atmospheric radiative balance. The results in Figure 2 show that the spatial patterns of ATM and SFC are similar, and they are also similar to those of Oikawa et al. (2013). Our study's results align with those of Matus et al. (2019), who reported a global mean aerosol direct radiative effect (DRE) of −2.40 W/m², primarily driven by sulfate aerosols with significant uncertainty due to aerosol type classification and optical depth retrievals. Similarly, our findings emphasize the critical role of accurate aerosol classification in determining the radiative forcing. Matus et al. (2019) also highlighted that anthropogenic aerosols contribute significantly to the global radiative effect, estimating an anthropogenic direct radiative forcing (DRF) of −0.50 W/m². Our study corroborates these findings, further illustrating the substantial impact of anthropogenic aerosols on the Earth's energy budget. Both studies underscore the value of leveraging satellite-based observations to capture aerosol radiative effects, particularly in regions where ground-based measurements are sparse. This can be attributed to the stronger influence of the underlying ground surface characteristics on the aerosol radiative forcing (Figure 2 (a), (d), and (g)). Aerosol radiative forcing plays an important role in the radiative balance of the atmosphere, and should be evaluated both at the surface and in the atmosphere. In the near future, the utilization of ATLID data from EarthCARE will be instrumental in enhancing the accuracy of aerosol radiative forcing assessments. For example, Nishizawa et al. (2019) demonstrated the potential of utilizing ATLID alone to estimate four additional aerosol components: WS, LA, DS, and SS. This expanded aerosol classification scheme

provides valuable insights into the composition and optical properties of aerosols, enabling a more detailed and accurate assessment of aerosol radiative forcing. By incorporating these additional aerosol components into RT simulations in future research, we can enhance our understanding of the impact of different species of aerosols on Earth's radiative balance.

Figure 3 shows the radiative forcing due to clouds, showcasing distinct characteristics in terms of SW and LW radiation and their effects at both the TOA and the SFC. The SW component of cloud radiative forcing leads to cooling effects at both TOA

and SFC, indicating that clouds reflect and scatter incoming solar radiation, reducing the amount of energy reaching the Earth's surface. Conversely, the LW component of cloud radiative forcing contributes to the heating effects at both the TOA and SFC as clouds absorb and re-emit LW radiation, trapping heat within the atmosphere. Furthermore, Figure 3 (d) shows that the SW component of the cloud radiative forcing generally induces heating within the ATM. This confirms that clouds have a warming effect on the atmospheric column by absorbing and redistributing the incoming solar radiation. In contrast, the LW component

of cloud radiative forcing displayed distinct patterns that varied with latitude. At low- and mid-latitudes, LW forcing contributes to heating, implying that clouds enhance downward thermal radiation, thus warming the atmosphere in these regions (Figure 3 (e)). However, at high latitudes, LW forcing results in cooling, indicating that clouds reduce the downward thermal radiation, leading to a net cooling effect. These characteristics also confirm the spatial variability of LW cloud radiative forcing at TOA, ATM, and SFC and its influence on the Earth's energy balance at different height levels, as found in previous

studies (e.g. Stephens et al. 2012). Our findings on cloud radiative forcing are consistent with those reported in previous studies, including Matus and L'Ecuyer (2017), L'Ecuyer et al. (2019), and Hang et al. (2019). These studies similarly identified significant impacts of clouds on radiative forcing at the top of the atmosphere, surface, and within the atmosphere, supporting the robustness of our results. Ham et al. (2017) compared CCCM with 2B-FLXHR-Lidar, showing regional differences in radiative fluxes due to differences in cloud characteristics within the products, and we believe that more detailed comparisons

between products, including our product, would be beneficial and needed to further improve the products as future work. EarthCARE, with its advanced cloud particle detection capabilities of ATLID and Doppler CPR, is expected to enhance the accuracy of cloud radiative forcing estimates. Understanding these spatial patterns sets the stage for a detailed evaluation of the algorithm's accuracy, which is discussed in the following sections through comparisons with CERES measurements.

**4. Comparison with CERES observation at TOA**

The radiative flux products were validated by comparing them with CERES observations at the TOA, which have a footprint of approximately 20 km, whereas CloudSat/CALIPSO provides information along a 1.8 km nadir path, making it difficult to make a perfectly consistent comparison. However, it is crucial to estimate errors through comparisons with observations, even when considering this discrepancy on a spatial scale. In this study, the results of the RT calculations were averaged over 20

325   km to closely match the CERES observations. In this section, a comparison between this product and CERES observations at the TOA is presented, and the verification and limitations of the errors under various cloud conditions are described and discussed. Section 4.1 exemplifies the vertical profiles of aerosols and clouds as crucial inputs to the current algorithm, along

with the results of radiative transfer calculations and comparisons with flux measurements. This section illustrates how aerosols and clouds can influence radiative transfer calculations, and how their measurement uncertainty can introduce radiative flux errors. In Section 4.2, the radiative fluxes between the RT simulations and observations are compared as monthly averages with a horizontal resolution of 5° to quantify the global bias and RMSE. The validity of this product was further confirmed by comparison with the radiation flux product (2B-FLXHR-Lidar) developed by the NASA CloudSat/CALIPSO team (https://www.cloudsat.cira.colostate.edu/). The radiative flux comparisons were then broken down into different thermodynamic phases of clouds, a factor that significantly impacts the radiative fluxes. Given that the flux error characteristics vary with the spatiotemporal scales, on which the assessment of aerosol and cloud radiative effects depends, the RMSE values were also investigated by altering the spatiotemporal scales (Section 4.3) to quantify the scale dependence of the RMSE in this algorithm.

## 4.1 Global validation and effects of cloud phase

In this section, radiative flux errors against CERES measurements are investigated on a global scale under different atmospheric conditions to assess the sources of flux errors in more detail. First, the global validation results are shown in the form of bias relative to CERES observations for both our product and NASA's product under all-sky conditions. Figure 4 shows comparisons between the calculated SW and LW radiative fluxes at the TOA and those from CERES observations at a spatial scale of 5 °and a temporal scale of 1 month. Each point in the scatter plots represents data from monthly 5 ° gridded points over four months in 2007. The SW flux in this study is positively biased relative to CERES (Figure 4 (a)), and the positive bias of 24.4 $Wm^{-2}$ is somewhat larger than that of 2B-FLXHR-Lidar R04/R05 (-1.2 $Wm^{-2}$ and -2.1 $Wm^{-2}$; Figure 4 (b)-(c)). However, the RMSE of this study (36.3 $Wm^{-2}$) is smaller than that of 2B-FLXHR-Lidar R04 (46.4 $Wm^{-2}$), indicating that the results of this study have less variability relative to CERES than does 2B-FLXHR-Lidar R04. Although our product shows a positive bias, the RMSE minus the bias, representing the variability component of the RMSE, tends to be smaller than that of the other products, as illustrated by the smaller scatter in Figure 4 (a) compared to Figures. 4 (b) and 4 (c). However, the LW fluxes in this study showed similar accuracy in terms of bias and RMSE as 2B-FLXHR-Lidar R04/R05 (Figure 4 (d)–(f)). Possible sources of positive bias in the SW flux can be attributed to flux errors found under a particular cloud thermodynamic phase condition, as discussed below.

The cloud thermodynamic phase has a significant impact on radiation; thus, its measurement uncertainty and treatment in RT simulations can introduce errors into the estimated radiative fluxes. Therefore, it is useful to classify the global flux comparisons described above into different cloud phases to assess the flux errors and identify the factors contributing to these errors. For this purpose, this study exploited cloud thermodynamic phase information obtained from CloudSat/CALIPSO (CC) and MODIS (MOD) to classify scenes into different cloud phase categories. Based on the phase characterization by CC, "Water" and "Ice" are defined such that all the vertical layers were composed of water (liquid) clouds and ice clouds, respectively, and "Mixed" is defined as a mixture of ice and water within the vertical profile (e.g., Matus and L'Ecuyer (2017)). For single-

layered clouds, these CC-based phase discriminations are combined with MOD-based binary phase discriminations of "Water" and "Ice" to produce the four phase categories of "Water/Water", "Water/Ice", "Ice/Water", and "Mixed" in the order of CC/MOD for the first three categories. The single-layer clouds are derived from CloudSat/CALIPSO, indicating cases where only one vertically continuous cloud layer was detected. The multi layered clouds as observed by CC, which are difficult to observe by MOD, are categorized separately as "Multi," and the discrimination between "Mixed" and "Multi" is based on the

cloud phase information obtained from CC.

   Figure 5 shows a breakdown of the comparison in Figure 4 according to cloud type with the different cloud phase conditions described above. Specifically, Figures 5 (a)–(d) and (f)–(i) show comparisons of single-layer clouds for the SW and LW fluxes, respectively, with varying cloud-phase characterizations by CC and MOD, while Figures 5 (e) and (j) show comparisons of multi-layer clouds for the SW and LW fluxes, respectively. When classifying cloud types, we use 1 km grid data and analyze

only cases where the entire approximately 20 km footprint of CERES along the track is covered by the same cloud type. To reduce flux errors arising from horizontal cloud heterogeneity, comparisons were performed only when clouds within the CERES footprint were of the same cloud type. When both CC and MOD indicate water clouds, the SW flux shows a slight negative bias, but both the bias (-11.7 $Wm^{-2}$) and RMSE (46.2 $Wm^{-2}$) are relatively small (Figure 5 (a)) compared to ice-containing clouds. On the other hand, the LW flux shows no substantial bias (0.8 $Wm^{-2}$) and a small RMSE (6.0 $Wm^{-2}$) (Figure

5 (f)). However, when ice clouds were included in the CC, the SW flux exhibited a positive bias and both the bias and RMSE were larger (Figure 5 (b)–(d)). Additionally, the LW flux exhibited a negative bias (Figure 5 (g)–(i)). The positive SW bias could have been caused by a possible overestimation of the ice cloud optical thickness obtained from MODIS, particularly given that the assumption of Voronoi-type ice particles is common among the radiative transfer simulation and the MODIS retrieval of ice cloud optical thickness. The positive bias was significantly reduced when the RT calculation was performed

with a 30% reduction in the ice cloud optical thickness, highlighting the key role of the ice cloud optical properties in determining the SW flux. Nakajima et al. (2019), who described the cloud property retrievals from shortwave reflectance, showed a COT bias of about 2.4 for ice clouds relative to MODIS products, so that a 30% reduction of COT for ice clouds with small COT can be considered reasonable. It was also confirmed that the positive bias was reduced when the RT calculation was performed by replacing the MODIS COT with the NASA product of MAC06S0 (not shown). The negative bias observed

in the LW for ice-containing clouds may be due to an overestimation of the cloud-top height obtained from the CC or an underestimation of the cloud-top temperature calculated by using the reanalysis data. The EarthCARE satellite is expected to reduce the LW bias by providing more accurate cloud detection through improved measurement instrumentation. Such advancements are expected particularly from technologies employed by the EarthCARE mission, which utilize improved instrumentation with higher spatial and spectral resolution, as well as enhanced sensitivity in detecting cloud properties,

especially those significant in the LW spectrum. For example, EarthCARE's advanced radar and lidar systems allow for more precise cloud profiling, which leads to more accurate detection and characterization of cloud cover and thickness. This improved accuracy in cloud detection helps reduce biases in LW radiative flux calculations by ensuring that cloud-related inputs to radiative transfer models are more representative of actual atmospheric conditions.  The same bias occurs in the case

of Multi as in the case of ice-containing clouds, which can be understood by the tendency for Multi to also contain a significant number of ice clouds, such as upper-level cirrus clouds.

## 4.2 Scale dependence of the flux error

Given the spatiotemporal scale dependence of radiative fluxes, validation of RT-simulated fluxes against measurements needs to be performed over different spatial and temporal scales. Such scale-dependent validation is also beneficial for product users who may use the data at various spatiotemporal scales depending on their analysis purposes. To this end, the RT-calculated and CERES-measured fluxes were compared at various spatiotemporal scales. Figure 6 shows a comparison of the variation in RMSE as a function of spatial and temporal scales for both all-sky SW and LW radiation fluxes at the TOA. All the results indicated a systematic reduction in the RMSE as the spatiotemporal scale increased. The reduction is more pronounced with changes in the spatial scale than with changes in the temporal scale. This can be attributed to the fact that the A-Train satellite constellation passes over the same location only twice a day, whereas a larger number of samples are available for comparison with observations at larger spatial scales, resulting in a smaller effect on the RMSE with an increasing temporal scale than with an increasing spatial scale.

The scale dependences of the flux error were also analyzed using the 2B-FLXHR-Lidar products and compared with those of our product, as shown in Figure 6. Compared to 2B-FLXHR-Lidar R04 and R05, this study exhibited lower RMSE values for SW radiative fluxes at lower spatial resolutions (Figure 6 (a)–(c)). At higher spatiotemporal scales, the RMSE was approximately 30 Wm$^{-2}$ at spatial and temporal scales of 40 °and 40 days, which was comparable to that of 2B-FLXHR-Lidar R04. However, for the 2B-FLXHR-Lidar R05 at high spatiotemporal scales, the RMSE decreased to approximately 10 Wm$^{-2}$ in contrast to our product, which had an RMSE of approximately 30 Wm$^{-2}$. This was attributed to a positive bias in the RT-calculated fluxes in this study relative to the observed values, as discussed in Sections 3.1 and 5.1. Regarding the LW radiation fluxes (Figure 6 (d)–(f)), the RMSE did not significantly decrease as the spatiotemporal scale increased in this study compared to 2B-FLXHR-Lidar R04 and R05. This is consistent with the negative bias of our product when ice clouds are included, as described in Section 5.1.

The scale dependence of the flux errors is also likely influenced by the cloud phase and type, which significantly influence the bias and RMSE, as demonstrated in Section 5.1. Given that the primary cloud types as sources of flux errors were identified based on the RMSE by classifying the cloud phases, the scale dependence of the flux error was also broken down into different cloud types. Figure 7 shows the RMSE statistics as a function of varying spatiotemporal scales for different cloud-type categories defined in the same manner as in Section 5.1. For the SW water cloud, the RMSE showed a decreasing trend as the spatiotemporal scale increased, reaching approximately 20 Wm$^{-2}$ (Figure 7 (a)). However, when ice clouds were included, the RMSE exceeded 80 Wm$^{-2}$ for smaller spatiotemporal scales, and remained at approximately 70 Wm$^{-2}$ for larger scales (Figure 7 (b)–(e)). Similarly, for LW radiation, the RMSE exhibited a decreasing tendency with increasing spatio-temporal scale in the case of water clouds. However, when ice clouds were included, the RMSE did not show a significant decrease, even at

larger spatiotemporal scales. The results from the cloud-type classification appear to explain the relatively small decrease in the all-sky RMSE with increasing spatiotemporal scale in Figure 6 (a) and (d), reflecting the small reduction in RMSE in ice-containing clouds at even larger scales. This suggests that the positive flux bias of ice-containing clouds demonstrated in Section 5.1 tends to exist across a varying range of spatiotemporal scales, consequently yielding a larger RMSE under all-sky

conditions.

## 5. Surface Validation

RT simulations can calculate surface (SFC) fluxes as well as TOA, and validating both TOA and SFC allows for a proper assessment of the impact of clouds and aerosols on atmospheric radiation. These findings highlight the importance of spaceborne active sensors in constraining surface and atmospheric fluxes, which are essential for accurate climate modeling.

However, without quantifying the uncertainties associated with these estimates, it is challenging to fully trust the information they provide. Therefore, the quantification of uncertainties is crucial to assess the reliability of the derived fluxes and their implications for climate science. The error quantification not only at TOA but also at SFC can also aid in assessing the influence of atmospheric radiation on ATM, enabling to characterize the flux errors in radiative energy partitioning into ATM and SFC. This section quantitatively evaluates the product by comparing instantaneous RT fluxes with ground-based BSRN observations.

Figure 8 compares the downward radiative fluxes calculated using the RT and those observed by the BSRN at the ground surface across different locations. In the comparison of SW radiation fluxes (Figure 8 (a), (c)), a minor bias of -17.6 $Wm^{-2}$ is noted, but the RMSE is substantial at 172.0 $Wm^{-2}$, with a sample number of 47. The discrepancies in the bias and RMSE values can be attributed to the limited number of samples available for comparison. Furthermore, the flux error may vary considerably depending on the specific measurement location, possibly owing to the potential issue of varying measurement

accuracy across different locations under different meteorological conditions, although no significant differences were observed between clear-sky and cloudy-sky conditions. In the comparison of LW radiation flux (Figure 8 (b), (d)), both the bias and RMSE (-5.6 and 19.0 $Wm^{-2}$) are significantly smaller compared to the SW flux, with a sample number of 44. The different number of samples between SW and LW reflects the variation in number of observation sites where SW and LW measurements are available. Unlike the SW radiation flux, no noticeable differences were observed in the calculated values

based on observation location. Additionally, no significant differences were observed between clear- and cloudy-sky conditions for the LW radiation flux. Assuming a scale dependence of the flux error similar to that of the TOA radiative flux described in Section 5, the bias and RMSE against the ground measurements are expected to decrease with increasing sample size. Compared to ground-based observations, CloudSat's narrow footprint of 1.8 km and the predominance of land-based observation sites make it challenging to expand the sample size. In future studies, the challenge will be to increase the sample

size by extending the period of RT simulations.

## 6. Conclusions

This paper describes the Level 2 (L2) algorithm for radiative flux and heating rate products generated by using the cloud and aerosol vertical information observed by CPR, ATLID, MSI, and BBR onboard the EarthCARE satellite. As a testbed before the launch of the EarthCARE satellite, A-Train data, particularly from CloudSat, CALIPSO, and MODIS, were used as inputs for the radiative transfer simulations, and the results were compared with CERES flux measurements at the TOA and with BSRN measurement at the surface. The aerosol and cloud radiative forcing derived from this product can be provided in the form partitioned into Top of the Atmosphere (TOA), Atmosphere (ATM), and Surface (SFC). This capability will serve to quantitatively assess the effects of aerosols and clouds on the Earth's energy budget. Comparisons with flux observations revealed good agreement for downward longwave (LW) radiation, but significant discrepancies were observed for downward shortwave (SW) radiation, particularly in the presence of clouds. The validation results showed that the SW radiation fluxes in this study exhibited a positive bias but had smaller variability than the previous 2B-FLXHR-Lidar products. The LW fluxes demonstrated a similar level of accuracy to previous products in terms of bias and root mean square error (RMSE). The analysis was broken down into different cloud types and phases, suggesting that the positive bias in SW fluxes is likely attributable to an overestimation of the optical thickness of ice clouds used as input. In addition, we quantified how the accuracy of radiative flux calculations varies with different spatial and temporal averaging scales. The increase in the scale led to a reduction in the RMSE and highlighted the importance of a larger sample size, particularly in the spatial dimension, to improve agreement with observations. Comparisons with ground-based observations from the BSRN showed a small bias in the SW radiative fluxes but an increased RMSE, potentially due to the limited number of ground-based measurements. These findings suggest the need for improved estimation of ice cloud properties to reduce the bias of SW radiative fluxes to achieve a more accurate radiation budget assessment and the cloud effect after the EarthCARE satellite is launched in 2024.

**Data availability.**

The MOD-Micro was provided by Prof. Takashi Y. Nakajima and Dr. Mirui Wang of Tokai University. Other JAXA products (CA-Aprop and CSCA-Micro) were obtained from the JAXA A-Train Product Monitor (https://www.eorc.jaxa.jp/EARTHCARE/research_product/ecare_monitor.html). The CloudSat data products of 2B-CWC-RO-R04 were provided by the CloudSat Data Processing Center at the CIRA/Colorado State University (https://www.cloudsat.cira.colostate.edu). MCD43C3 cells were obtained from the LP DAAC website (https://lpdaac.usgs.gov/products/mcd43c3v006/). CERES and GEOS-4,5 data included in the CCCM were obtained from NASA/LARC/SD/ASDC (https://doi.org/10.5067/AQUA/CERES/CCCM-FM3-MODIS-CAL-CS_L2.RELD1).

**Author contributions.**

AY performed the data analysis and the radiative transfer simulations. KS coordinated this work and obtained a funding. AY and KS produced the final manuscript draft. All authors reviewed the manuscript. MS, EJ, and HI developed the basic design

of the radiation code. MS created Voronoi non-spherical tables. TMN provided feedback on the analysis methods as well as on the manuscript draft.

**Competing interests.**

The authors have no competing interests to declare.

**Disclaimer.**

Publisher's note: Copernicus Publications remains neutral with regard to jurisdictional claims in published maps and institutional affiliations.

**Special issue statement.**

This article is part of the special issue "EarthCARE Level 2 algorithms and data products." It is not associated with a conference.

**Acknowledgements.**

The authors would like to thank the members of the JAXA EarthCARE Science Team for their support and advice. The
505 authors are also grateful to four anonymous reviewers for their invaluable comments that greatly improved this paper. The authors would also like to thank Editage (www.editage.jp) for English language editing.

**Financial support.**

This study was funded by the Japanese Aerospace Exploration Agency (JAXA) EarthCARE Project (grant no.
24RT000226).

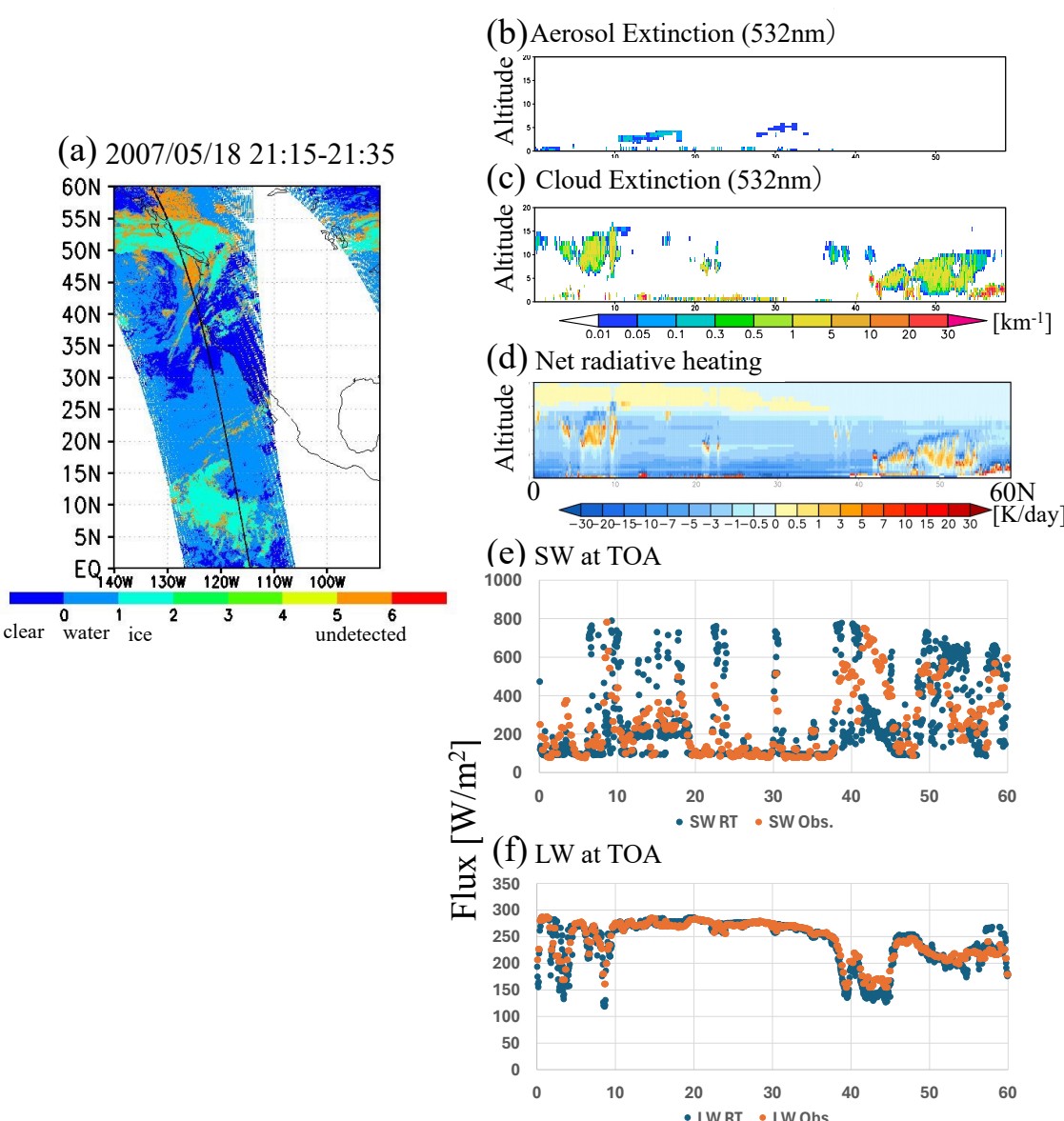

Figure 1.  Inputs and outputs from the flux algorithm, (a) Footprint of the A-Train is represented by the solid line, with cloud phase information from MODIS indicated in color, (b) input data for Aerosol Extinction, (c) Cloud Extinction (in km⁻¹), and (d) Net radiative heating (in K/day). (e) Comparison of RT calculations for SW RT calculations (blue) with CERES observations (orange) at the TOA. (f) Same as (e) but for LW at the TOA.

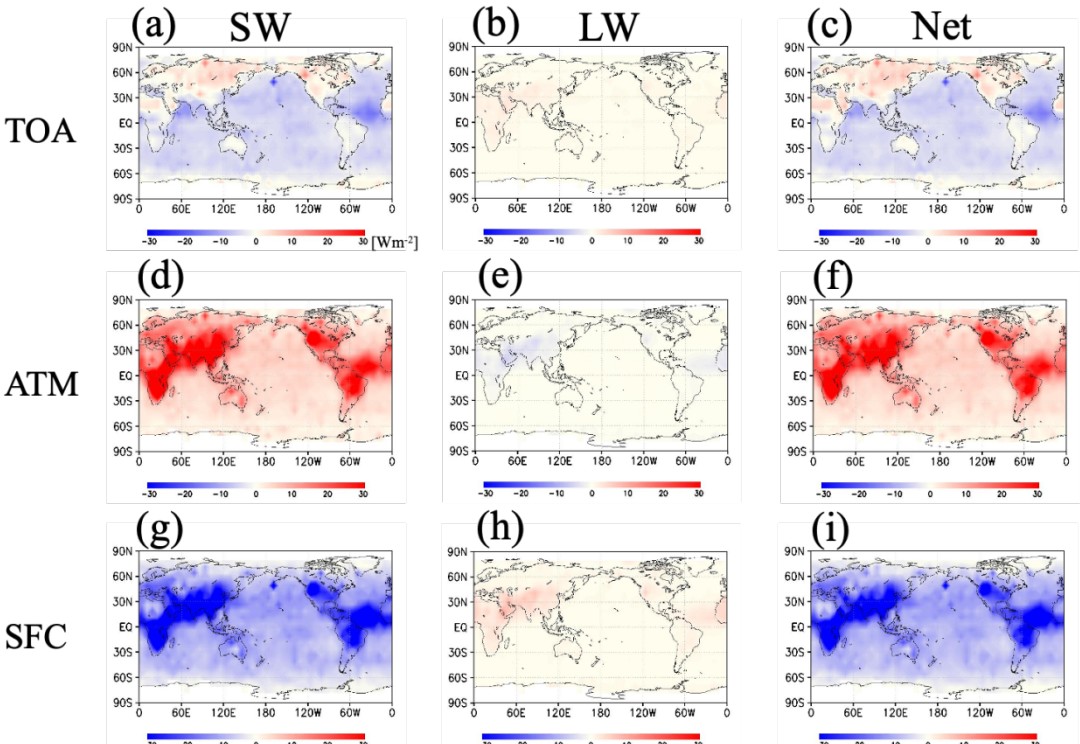

Figure 2. The 4-month average SW, LW, and net aerosol radiative effects under all-sky conditions. In (a), (d), and (g), the focus is on SW effects, and (b), (e), and (h) provide insights into LW effects. (c), (f), and (i) depict Net effects, each corresponding to different atmospheric levels: (a-c) TOA, (d-f) ATM, and (g-i) SFC. Color units are in Wm$^{-2}$.

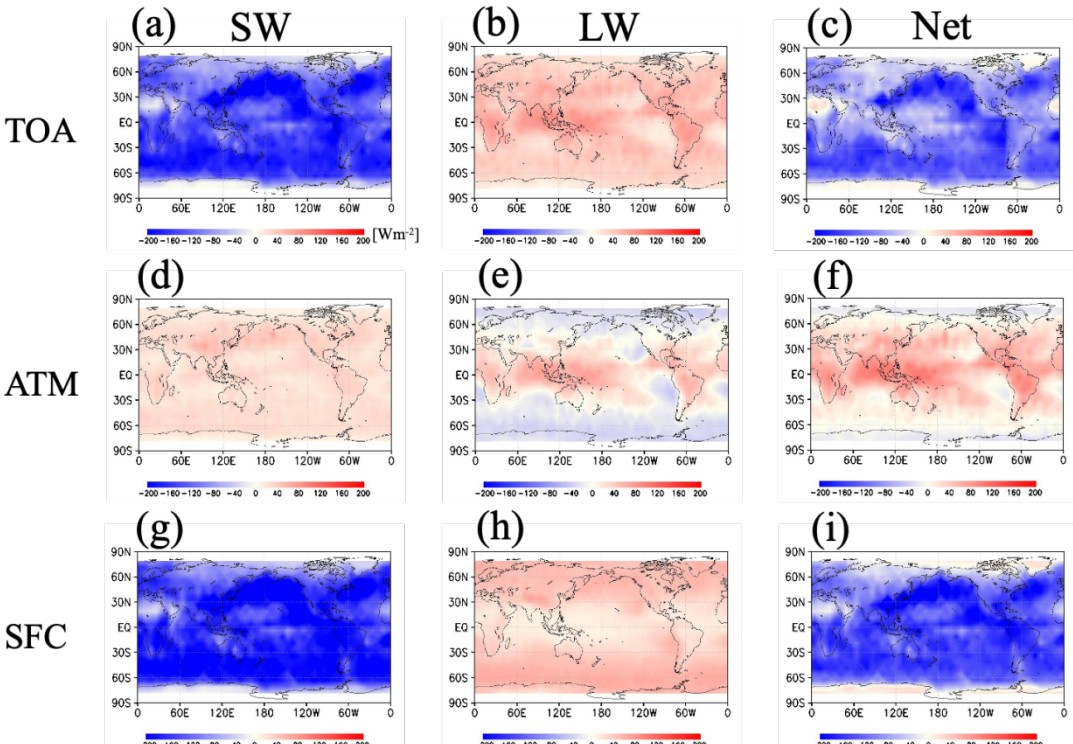

Figure 3. Same as Figure 2 but for cloud radiative effects. Color units are in Wm⁻².

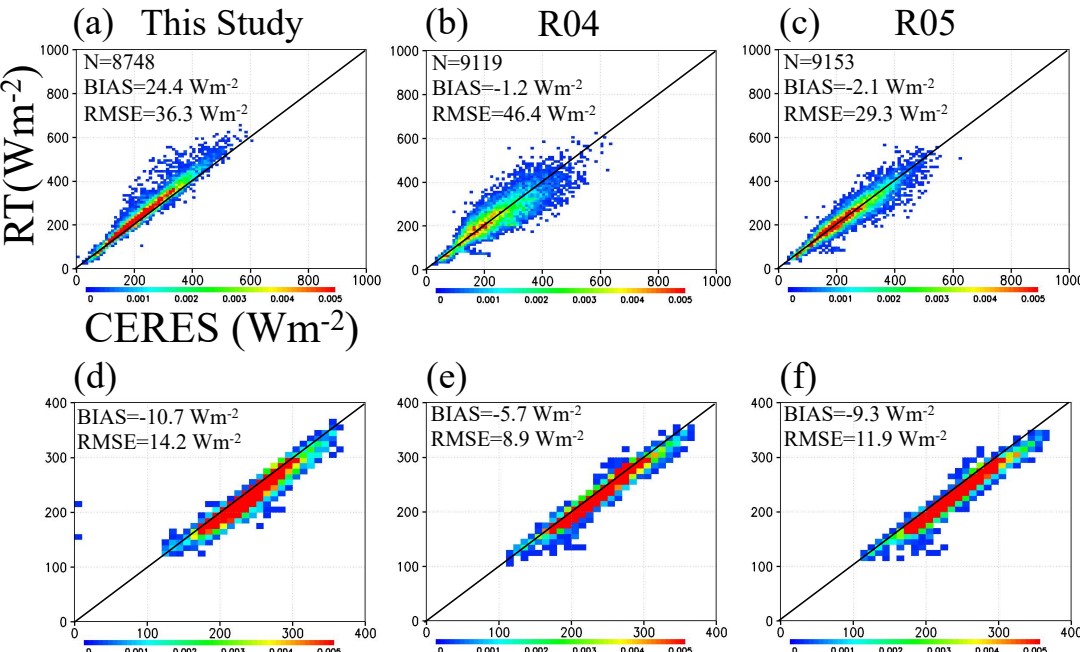

Figure 4. Comparison of monthly, 5°mean TOA flux RT calculations and CERES observations. (a.d) This study, (b,e) 2B-FLXHR-Lidar R04, and (c,f) 2B-FLXHR-Lidar R05, (a)-(c) out going SW radiation, and (d)-(f) out going LW radiation. The color represents frequency.

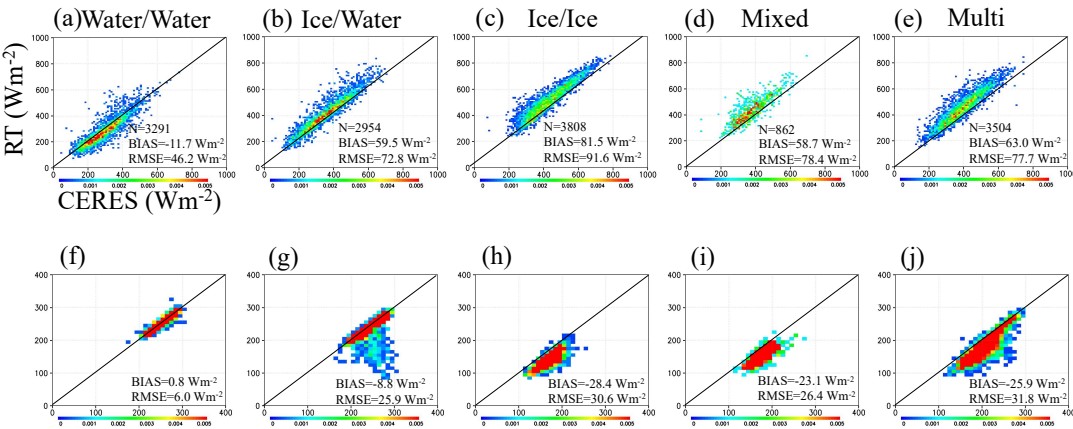

Figure 5. Same as Figure 4 (a) and (d), but classified by cloud type. (a,f) water (CC)/water (MOD), (b,g) ice (CC)/water (MOD), (c,h) ice (CC)/ice (MOD), (d,i) mixed phase (CC), and (e,j) multi layered (CC) clouds. CC indicates cloud phase from

CloudSat/CALIPSO and MOD indicates cloud phase from MODIS. (a)-(e) out going SW radiation, and (f)-(j) out going LW
radiation. The color represents frequency.

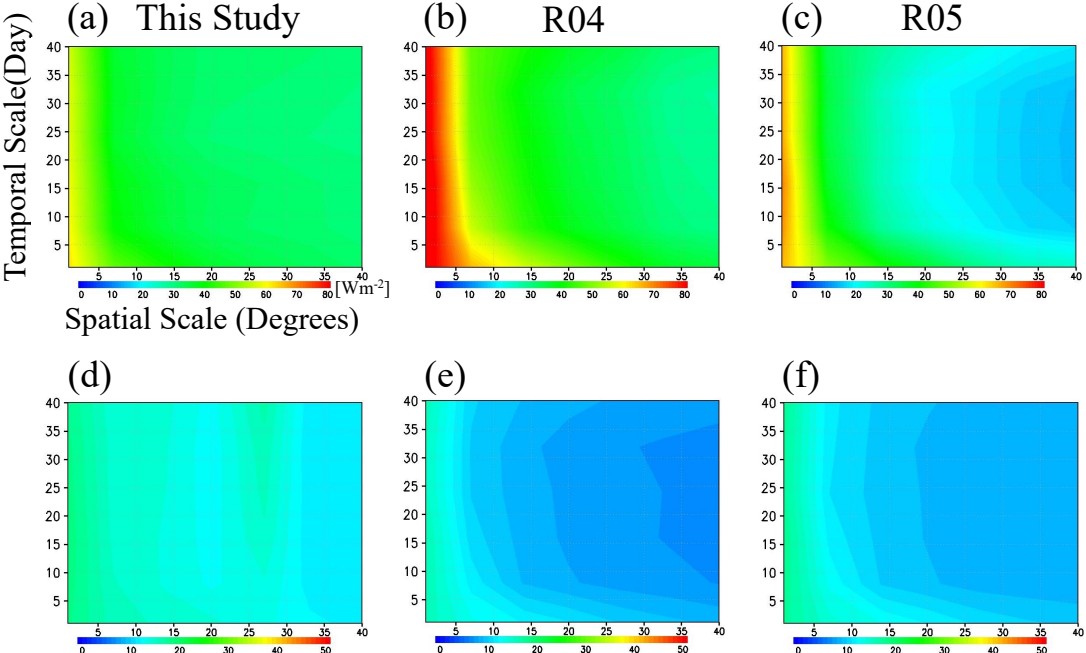

Figure 6. RMSE differences between TOA flux RT calculations and CERES observation on a variety of time and space scale.
(a.d) This study, (b,e) 2B-FLXHR-Lidar R04, and (c,f) 2B-FLXHR-Lidar R05, (a)-(c) out going SW radiation, and (d)-(f) out
going LW radiation. Color units are in Wm⁻².

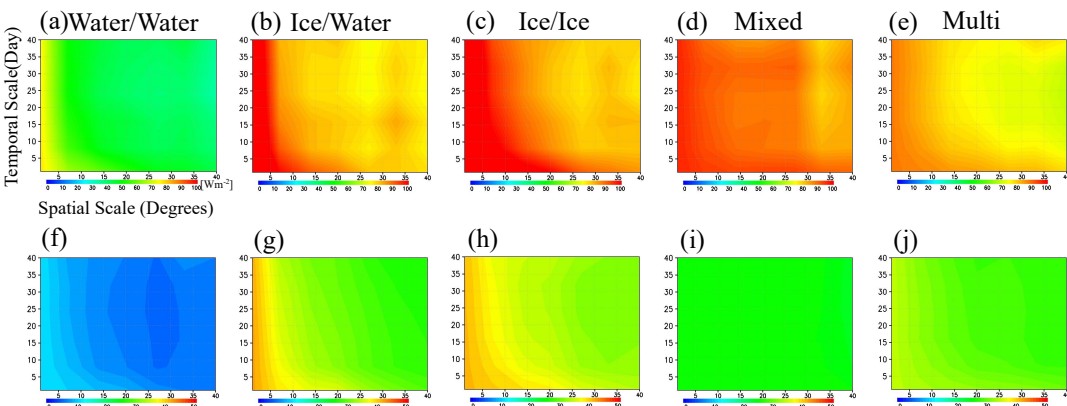

Figure 7. Same as Figure 6 (a) and (d), but classified by cloud type. (a,f) water (CC)/water (MOD), (b,g) ice (CC)/water (MOD), (c,h) ice (CC)/ice (MOD), (d,i) mixed phase (CC), and (e,j) multi layered (CC) clouds. CC indicates cloud phase from CloudSat/CALIPSO and MOD indicates cloud phase from MODIS. (a)-(e) out going SW radiation, and (f)-(j) out going LW radiation. Color units are in Wm$^{-2}$.

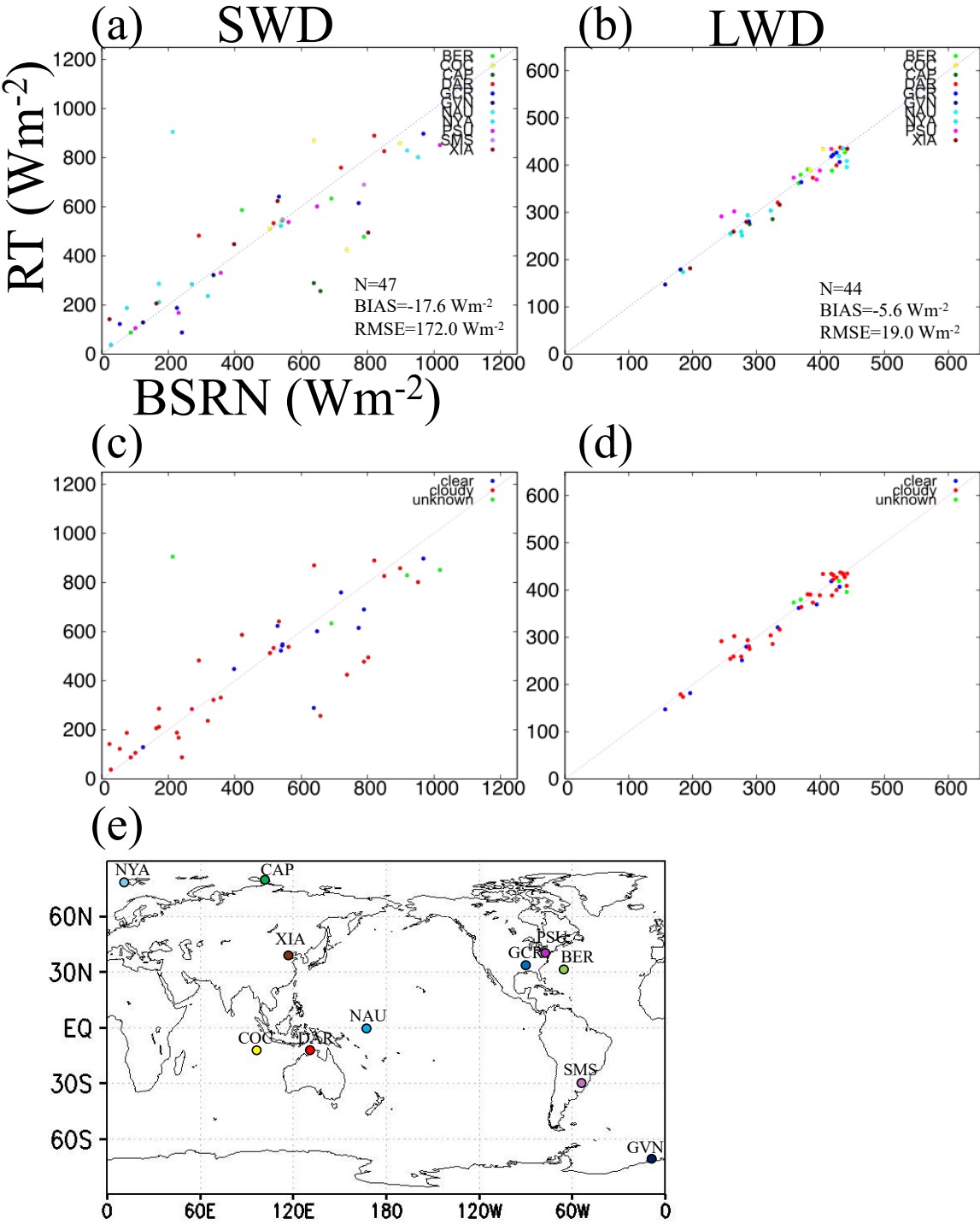

Figure 8. Comparative analysis between instantaneous surface flux RT calculations and observations obtained from the BSRN. (a) and (b) illustrate the data points plotted according to observation locations, (c) and (d) categorize atmospheric conditions (blue: clear, red: cloudy, green: unknown). In detail, (a) and (c) focus on surface SW radiation, and (b) and (d) emphasize surface LW radiation. (e) A global map of location for BSRN observation sites. The names of BSRN stations are as follows: BER (Bermuda), COC (Cocos Island), CAP (Cape Baranova), DAR (Darwin), GCR (Goodwin Creek), GVN (Georg von Neumayer), NAU (Nauru Island), NYA (Ny-Ålesund), PSU (Rock Springs), SMS (São Martinho da Serra), and XIA (Xianghe).

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
