# Peer review of "Description and validation of the Japanese algorithm for radiative flux and heating rate products with all four EarthCARE instruments: Pre-launch test with A-Train"

_Atmospheric Measurement Techniques, 2024_

## Referee Comment (RC4)

**Review**

This manuscript describes the theoretical foundations of the Japanese radiative flux and heating rates product for EarthCARE. The algorithm derives vertical profiles of longwave (LW) and shortwave (SW) radiative fluxes and heating rates at 34 atmospheric levels by applying a radiative transfer model to aerosol and cloud profiles retrieved from the EarthCARE cloud profiling radar, lidar, and multi-spectral imager. The primary focus of this study is to document the anticipated accuracy of the product by applying the algorithm to existing observations collected by the A-Train. The subject is appropriate for *Atmospheric Measurement Techniques* and the uncertainty analysis is quite thorough considering the algorithm has yet to be implemented for EarthCARE. My primary concerns center on the organization of the findings. In particular, the abrupt transition from the algorithm description to validation could be softened by including the preliminary results prior to discussing the comparisons. In addition, there are several opportunities to reference related literature that should be considered. Since I do not anticipate those modifications requiring substantial rewriting, I recommend the paper be published in AMT after the following minor revisions to address these concerns.

Specific Comments:

1. The most significant issue with the paper in its current form is the organization of results. This transition from algorithm description immediately into comparisons with CERES is quite abrupt. It would be interesting to see some examples of the algorithm before discussing its evaluation. I think the example in Figure 1 could be used to simply illustrate the methods described in Section 2 (omitting the CERES comparisons in panel (e) which are hard to see anyway). That could be followed the spatial distributions of aerosol and cloud radiative effects in Figures 6 and 7 to provide context for what the algorithm does before assessing the accuracy of these results.

2. Line 43: The acronym for CERES is missing some words "Clouds and the Earth's Radiant Energy System"

3. Line 49: Since this is not the first paper to estimate fluxes using radiative transfer modeling with atmospheric inputs, I suggest referencing some of the pioneering papers on this topic (e.g. Rossow and Lacis, 1990; Rossow and Zhang, 1995; Zhang et al, 1995; Whitlock et al, 1995).

4. Line 69: It may also be worth adding that these measurements will provide important continuity for the data record that began with the A-Train in 2006 (L'Ecuyer and Jiang, 2010).

5. Line 90: While it is likely beyond the scope of this particular study, there could be value in digging deeper into comparisons with FLXHR-lidar and CCCM to trace the source of discrepancies in all three algorithms. Since the algorithm has already been applied to CloudSat/CALIPSO/MODIS observations, it could immediately be compared to FLXHR-lidar and CCCM in a manner like that of Ham et al. (2014). The results would be very interesting for understanding all three algorithms.

6. Line 113: I think 'were utilized' should be 'will be utilized' since EarthCARE data were not actually used in this paper.

7. Line 157: Do you mean 'daytime' instead of 'diurnal'?

8. Line 180 (and again on Line 311): The spatial resolution of CloudSat is 1.4 km (across track) by 1.8 km (along track).

9. Line 229 - 231: There is precedence for separating results according to cloud phase in this way. Perhaps cite Matus et al. (2017) here.
10. Line 297: The preceding discussion does not provide adequate context for the value of these estimates. The ability of spaceborne active sensors to constrain surface fluxes and atmospheric flux divergence represents one of the most important contributions they have made to climate science. This is discussed in detail in papers like Haynes et al. (2010), L'Ecuyer et al. (2019), and Hang et al. (2019), for example. If this is better articulated in the introduction, the point here could be that without quantifying the uncertainties, it is hard to know how trustworthy this information is.
11. Line 328: It would also be good to compare against other recent studies that produce similar estimates (Matus et al, 2019 is one example but there are others, including some by Winker et al.)
12. Line 337: Similarly, some qualitative comparisons against prior work are warranted here as well (there are lots of options but Matus et al, 2017; L'Ecuyer et al, 2019 and Hang et al, 2019 all utilize similar observations to extract the effects clouds at TOA, SFC, and in the ATM).
13. Line 363: This isn't an accurate statement. The analysis quantifies how the accuracy of radiative flux calculations varies with spatial and temporal averaging scale.
14. Figure 1: The transition from yellow to light blue in the upper atmosphere in Figure (d) is likely an artifact of the color bar. It might be good to have a small band of white from -0.05 to 0.05 to represent areas of 0 heating.
15. Figure 3 caption: Technically this figure is only the same as Figure 2 panels (a) and (d).
16. Figure 5 caption: Again, this figure is only the same as Figure 4 panels (a) and (d).
17. There are also several minor grammatical errors throughout the paper. A few representative examples follow, but I suggest taking a careful read through the paper for other similar issues:
    a. Line 38: 'circulation' should be 'circulations'
    b. Line 44: 'radiometer' should be 'radiometers'
    c. Line 199: 'value' should be 'values'
    d. Line 200: 'of the aerosols' should be 'of aerosols'

References

1. Ham, S.-H., S. Kato, F. G. Rose, D. Winker, T. L'Ecuyer, G. G. Mace, D. Painemal, S. Sun-Mack, Y. Chen, and W. F. Miller, 2017: Cloud occurrences and cloud radiative effects (CREs) from CCCM and CloudSat radar-lidar products, *J. Geophys. Res.* **122**, 8852-8884.
2. Hang, Y., T. S. L'Ecuyer, D. Henderson, A. V. Matus[s], and Z. Wang, 2019: Reassessing the role of cloud type in Earth's radiation budget after a decade of active spaceborne observations. Part II: Atmospheric heating, *J. Climate* **32**, 6219-6236.
3. Haynes, J. M., T. H. Vonder Haar, T. L'Ecuyer, and D. Henderson, 2013. Radiative heating characteristics of Earth's cloudy atmosphere from vertically resolved active sensors, *Geophys. Res. Letters* **40**, doi:10.1002/grl.50145.
4. L'Ecuyer, T. S. and J. Jiang, 2010: Touring the Atmosphere Aboard the A-Train, *Physics Today* **63** (7), 36-41.
5. T. S. L'Ecuyer, Hang, Y., A. V. Matus, and Z. Wang, 2019: Reassessing the role of cloud

type in Earth's radiation budget after a decade of active spaceborne observations. Part I: Top of atmosphere and surface, *J. Climate* **32**, 6197-6217.

6. Matus, A. and T. S. L'Ecuyer, 2017: The role of cloud phase in Earth's radiation budget, *J. Geophys. Res.* **122**, doi:10.1002/2016JD025951.

7. Matus, A. V., T. S. L'Ecuyer, D. S. Henderson, and T. Takemura, 2019: New global estimates of aerosol direct radiative effects, kernels, and forcing, from active satellite observations, *Geophys. Res. Letters* **46**, 8338-8346.

8. Rossow, W. B., and A. A. Lacis, 1990: Global, seasonal cloud variations from satellite radiance measurements. Part II: Cloud properties and radiative effects, *J. Clim.*, **3**, 1204–1253.

9. Rossow, W. B., and Y.-C. Zhang, 1995: Calculation of surface and top of the atmosphere radiative fluxes from physical quantities based on ISCCP data sets: 1. Validation and first results, *J. Geophys. Res.*, **100**, 1167–1197.

10. Whitlock, C. H., and Coauthors, 1995: First Global WCRP Shortwave Surface Radiation Budget Dataset. *Bull. Amer. Meteor. Soc.*, **76**, 905–922, https://doi.org/10.1175/1520-0477(1995)076<0905:FGWSSR>2.0.CO;2.

11. Zhang, Y.-C., W. B. Rossow, and A. A. Lacis, 1995: Calculation of surface and top of the atmosphere radiative fluxes from physical quantities based on ISCCP data sets: 1. Method and sensitivity to input data uncertainties, *J. Geophys. Res.*, **100**, 1149–1165.

---

## Author Comment (AC1)

**Response to Referee #1**

Thank you very much for your time and effort taken to review our manuscript submitted to AMT. We really appreciate the reviewer's constructive comments that are very useful to greatly improve the manuscript. We have revised the manuscript based on your comments as explained below. Please see below for our point-by-point responses to your comments, where the original comments are shown in *italics* and our responses are shown in normal text just below your corresponding comments.

*General comments*

1. *The evaluation of the algorithms using A-Train are useful but it would be helpful if the text could be restructured so that it is clearer what inputs will be used as input when EarthCARE data are available and what is being used from A-Train. For example, in Section 2.1 there is a mixture of EarthCARE products and A-Train data which is a bit challenging to parse.*

A. To clarify the distinction between the inputs used from A-Train data and those that will be used from EarthCARE products, we have added the following text at the beginning of Section 2.2:

'In the analysis of this paper, we utilize data from the A-Train constellation at the time of writing this paper before the EarthCARE data becomes available. While EarthCARE products will be used for future operational applications, A-Train data, including observations from CloudSat, CALIPSO, and MODIS, are currently employed to evaluate and refine the algorithm in preparation for application to the EarthCARE data. The A-Train data provides a valuable proxy for the type of information that will be available from EarthCARE, although there are some differences in instrument characteristics and data resolution. These differences are taken into account in our analysis to ensure that the results are relevant for the upcoming EarthCARE mission. For the EarthCARE mission, the algorithm will utilize data from the CPR_CLP (from CPR), the ATL_CLA (from ATLID), and the MSI_CLP (from MSI). These instruments provide vertical profiles of clouds and aerosols, which are critical inputs for calculating radiative fluxes and heating rates. The A-Train data, on the other hand, allows us to test and validate the algorithm using observations that are similar in nature to those expected from EarthCARE, ensuring that the algorithm is robust and ready for operational use once EarthCARE data becomes available.'

2. *In the manuscript evaluation of the RT products are compared with different datasets. It is easier to keep track of things if the products are introduced then consistently referred to after that point.*

3. *The methodology used for the TOA flux evaluation, Section 3.2, is difficult to understand in detail, especially the analysis of fluxes by cloud phase. Detailed comments have been provided below.*

A. The response to the general comments 2 and 3 will be addressed through the replies provided in each of the detailed comments below. We have added explanations, included citation of additional papers, and revised sentences to make the text clearer in accordance with the points raised in the comments.

*Detailed comments*

*Line 47-48: This sentence should be rewritten to make it clearer that you are discussing space-based estimates of in atmosphere and surface radiative fluxes. The current text is a bit confusing, at least to me, since I wondered why RT calculations were used for surface radiation fluxes instead of surface-based radiometers.*

A. The following statement has been added to clarify the explanation.
  'Space-based RT calculations are commonly used to estimate radiative fluxes within the atmosphere and at the surface, complementing surface-based measurements where direct observations may be limited or unavailable on the global scale.'

*Line 75:   References for EarthCARE instruments or products?*

A. We have cited Illingworth et al. (2015) and Wehr et al. (2023) as references for EarthCARE instruments and products in the revised manuscript.

*Line 86: The +/- 10 W/m^2 has a particular spatial scale (averaged over 100 km^2) and temporal scale (instantaneous) which should be noted or referenced here.*

A. References (ESA, 2001) and the following explanations have been added.

"The uncertainty of $\pm 10$ Wm$^{-2}$ is associated with a spatial scale averaged over 100 km² and is based on instantaneous BBR measurements."

*Line 126: Is this total cloud water content (liquid + ice) or liquid water content?   If it is the former, how is it parsed into liquid and ice?*

A. This intends to mean 'liquid water content'. We have revised the text as 'the effective particle radius and Cloud Water Content (CWC) for liquid clouds.

*Line 140:   Why is a constant sea surface albedo used?*

A. A constant sea surface albedo is used in this study for simplicity and to reduce computational complexity. This approach assumes that the variability in sea surface albedo has a minimal impact on uncertainty of the overall radiative flux calculations, especially in comparison to other factors such as cloud cover and aerosol concentration. We have added a brief explanation of this setting in the revised manuscript as follows:

'In this study, a constant sea surface albedo is used to simplify the radiative transfer calculations and to minimize computational demands. This assumption is based on the understanding that variations in sea surface albedo have a relatively minor effect on uncertainty of the overall radiative flux compared to other variables such as cloud cover and aerosol properties.'

*Line 142: Significantly more detail is required about why the data was averaged, how the averaging was performed and the effect of the averaging on the resulting radiative transfer.   Examples of some questions that should be addressed include:*

   1.   *Why is the data not on the target 1 km along track grid for EarthCARE?*

   A.   The 1 km grid calculation has not been implemented in order to achieve operational computational speed. Furthermore, the current grid resolution has been chosen to align with the footprint of instruments like BBR and CERES, which is around 10 km to 20 km, respectively. However, we are considering to perform the calculations on a 1 km grid as part of future research products. We have added these explanations in the revised

manuscript as: "This averaging is primarily due to the computational cost of radiative transfer for meeting the latency requirement of data processing and is also for consistency with the footprint of BBR and CERES, which is around 10km and 20km, respectively."

2. *What is the original resolution of the individual datasets (cloud, aerosol, surface and meteorological fields)?*

A. The original resolution of all the individual datasets, including cloud, aerosol, surface, and meteorological fields, is 1 km × 240 m. MODIS global albedo product (MCD43C3) is gridded at a 0.05° by 0.05° spatial resolution. This information was added in the revised manuscript.

3. *If I assume that the retrieved cloud profiles are meant to represent ~ 1 km footprint (line 180), how were the cloud properties averaged in the horizontal? Are the resulting cloud profiles on the 5 km grid assumed to be overcast and horizontally homogeneous or instead partial cloud (cloud fraction < 100%) and inhomogeneous?*

A. If even a single grid within the 5 km grid contains clouds, the cloud profile for the entire 5 km grid is treated as uniformly cloudy, with values averaged horizontally. The original product is designed with a 1 km footprint resolution, but the 5 km grid assumes horizontal uniformity of cloud distribution within the grid, and values are averaged accordingly to account for any inhomogeneity. This explanation was added in the revised manuscript.

4. *How was the data averaged in the vertical?*

A. The vertical resolution of the radiation transfer model is 1 km from the Earth surface to 30 km altitude. The data was averaged onto the 1km resolution.

*Line 151: Is the Voronoi ice particle shape consistent with the EarthCARE retrievals?*

A. The particle shape of the Voronoi ice is consistent with assumptions in MSI cloud retrievals of EarthCARE. We have added the reference (Wang et al. 2023) to this and revised the text as follows.

'As an assumption of the ice cloud optical properties, Voronoi particles were used to account for the non-spherical shape of the ice particles in both the JAXA/A-Train product and the

EarthCARE mission (Wang et al., 2023). This assumption of ice particles in the RT simulation was consistent with that of the MODIS and MSI ice cloud retrievals.'

*Line 154: The CERES product and its version that was used for evaluation should be specified.*

A. We have added the specific product name, CER_ES8_Aqua-FM3_Edition3, to the manuscript to clarify the source of the CERES data used in our study.

*Line 157:  The method used to compute the diurnal fluxes should be explained.  For example, is there a consistent method used for the CERES and the 2B-FLXHR-Lidar algorithms.  Is the data in the product diurnal fluxes or instantaneous?  Are diurnal fluxes computed for comparison with BSRN data?  How was that done with the calculations and with the BSRN data?*

A. All comparisons, including those with other products and BSRN data, were conducted using instantaneous data. Diurnal fluxes were not computed or used in this study. This comparison is consistent with the CERES and 2B-FLXHR-Lidar algorithms. The use of instantaneous values for comparisons is noted in the revised text as follows.

'All comparisons, including those with other products and BSRN data, were conducted using instantaneous data.'

*Line 160: Is the analysis split to periods when MODIS was and was not available?  This affects the availability of the COT constraint on the cloud properties.*

A. All of the data were analyzed for the period over which MODIS was available. The data sampling for comparisons of the all-sky conditions includes the case with MODIS data not available, but the analysis for cloud phase type classification is based on the cases with MODIS data available. We hope the original text describing this now makes sense for you.

*Line 165: When averaging the RT results, they are an average of 20 km along orbit? I assume the CERES footprint is not just along orbit but roughly a 20x20 km footprint.*

A. The CERES flux data is 20 km x 20 km including both along-track and cross-track directions; however, the 1D radiative transfer calculation compares the flux calculated only in the along-track direction, so the comparison with CERES requires consideration of this point. This statement has been added to the text as follows in the revised manuscript.

'The CERES flux data is 20 km x 20 km including both along-track and cross-track directions; however, the 1D radiative transfer calculation compares the flux calculated only in the along-track direction, so the comparison with CERES requires consideration of this point.'

*Line 176: Please indicate the value of heat content of air at constant pressure used in the calculation.*

A. The specific heat content of air at constant pressure used in the calculation is $c_p$=1005 J kg$^{-1}$ K$^{-1}$. We have added this number in the revised manuscript.

*Line 190: It would be clearer to refer to products used for comparison after they have been introduced earlier in the text. It is not clear what data is "the NASA CloudSat CALIPSO team".*

A. We have specified the data product of 2B-FLXHR-Lidar for clarification and added the URL of the NASA team's website.

*Line 195: Maybe more precise to call it "cloud top phase of MODIS"?*

A. Corrected to "cloud top phase of MODIS".

*Line 202: What is the latitude resolution of the data shown in Figure 1 b-e? Is it 5 km?*

A. Yes, it is 5km. We have added this information in the revised manuscript as follows.

'The latitudinal resolution in panels (b) to (e) of Figure 1 is shown at 5 km.'

*Line 217:   The 24.4 W/m^2 bias is significantly larger than 2B-FLXHR-Lidar.*

A. The reason for the large bias is due to the positive bias in the case of ice-phase cloud. The positive bias due to ice-phase clouds is discussed in the section where comparisons are performed for different cloud phases separately (Section 4.1).

*Line 225: It would be good to indicate here the fraction of the full set of RT calculation that are used for the cloud type analysis. The text in this paragraph suggests that only data for which CloudSat, CALIPSO and MODIS are available will be used. As noted in 160, when the MODIS COT is not available that constraint is removed from the cloud properties used for the RT calculations.*

A. Although the percentage of occurrence is different depending on cloud type, comparisons with CERES are made on a 5° monthly average and are therefore presented as a sample size of N.

*Line 229-235: The categories are confusing, at least to me, and I suggest some restructuring and rewriting of the text to try and clarify them.   Summarizing my understanding of the current text, cloud phase based on CloudSat/CALIPSO data is "Water" when all layers are liquid phase, "Ice" when all layers are ice phase and "Mixed" when both are present.   However, only for single layer clouds is the combined CloudSat/CALIPSO and MODIS cloud phase categories defined.   This results in the categories "Water/Water", "Water/Ice", "Ice/Water".   Are the cloud phase categories unique? It is also not quite clear what is a single layer for the analysis.   Is it a single Cloudsat/CALIPSO layer or it can be multiple adjacent layers?*

A. The reviewer's understanding is correct for the cloud phase categories based on CloudSat/CALIPSO (CC), which generates "water", "ice" and "mixed". For the single-layer

clouds, these CC-based cloud phase categories are further combined with MODIS-based cloud phase categories of "water" and "ice" to result in combined categories of "water/water", "water/ice" , "ice/water" (in the order of CC/MODIS) and "mixed", as described in the text. These four phase categories are determined uniquely for a given single-layer cloud. Additionally, because it is challenging for MODIS to capture multi-layer clouds, our analysis with the CC-MODIS combined cloud phase information focuses on single-layer clouds. The single-layer clouds are derived from CloudSat/CALIPSO, indicating cases where only one vertically continuous cloud layer was detected. To clarify this point, the following sentence has been added to the revised manuscript.

'The single-layer clouds are derived from CloudSat/CALIPSO, indicating cases where only one vertically continuous cloud layer was detected.'

*Summing the "N" values in the Figure 3a, 3b and 3c, does not result in a total "N" that matches "N" shown in Figure 2a so it is not clear if the categories are unique.*

A. Figures 3(a)-(e) are derived from the classification and analysis of Figure 2(a), but since each cloud type is compared with CERES on a 5° monthly average basis, the sample sizes do not match exactly.

*Line 234: No need to restate MODIS(MOD) since it is done in line 229.*

A. MODIS (MOD) was corrected to MOD.

*Line 237: How is the "Mixed" category a single layer cloud when it is defined as "a mixture of ice and water within the vertical profile"?   This goes back to the comment about definition of a single cloud layer.*

A. "Mixed" indicates cases where both liquid water and ice were detected within the vertical structure obtained from CloudSat/CALIPSO.

*Line 239: It is stated that Figure 3 is the same as Figure 2 but broken down by cloud phase. This can be taken to mean that the data used to construct Figure 3 are derived from data averaged over 5 degrees and 1 month. If this assumption is correct, then it is unclear how to interpret the statement that the comparisons was limited to points when cloud in the CERES footprint were of the same type since that occurs on ~20 km and instantaneous data. When accumulated over space and time wouldn't there be heterogeneity arising from the CERES footprint level data, even if it was the same cloud type?*

A. When classifying cloud types, we use 1 km grid data and analyze only cases where the entire approximately 20 km footprint of CERES along the track is covered by the same cloud type. It is true that the CERES footprint also has a 20 km observation width in the cross-track direction, meaning that other types of clouds could be mixed in. However, our approach does not include these cases for simplicity, and this is considered a limitation of the current analysis. The following explanatory text was added.

'When classifying cloud types, we use 1 km grid data and analyze only cases where the entire approximately 20 km footprint of CERES along the track is covered by the same cloud type.'

*Line 243: Compared to what are the bias and RMSE are relatively small? While not necessary to include in the paper, it would be helpful to have the cloud phase analysis applied to the 2B-FLXHR-Lidar product to provide a point of comparison results using the EarthCARE algorithm.*

A. The RMSE is smaller than that of ice-containing clouds. We have added the following text in the revised manuscript to clarify this point: 'When both CC and MOD indicate water clouds, the SW flux shows a slight negative bias, but both the bias ($-11.7$ $Wm^{-2}$) and RMSE ($46.2$ $Wm^{-2}$) are relatively small (Figure 5 (a)) compared to ice-containing clouds.' This paper focuses on the validation of the Japanese product, and therefore, classifying and analyzing the 2B-FLXHR-Lidar data by cloud type is beyond the scope of this validation. However, scientifically, it is very meaningful to validate the 2B-FLXHR-Lidar data by cloud type and to compare with our product, and we plan to do so in future EarthCARE validation studies.

*Line 312:   Could the biases also be compared with computed surface fluxes from CERES and 2B-FLXHR-Lidar?   While not direct observations they would increase the amount of data that could be used for comparison with the EarthCARE RT algorithm.*

A. As part of the EarthCARE validation plan, it has been decided to use observed flux data from BBR and BSRN, so we have conducted comparisons with these observations. In this paper, the validation was performed solely by comparing with observational data, in accordance with the validation plan. However, comparisons with surface fluxes from CERES and 2B-FLXHR-Lidar, suggested by the reviewer, would also be valuable to further validate our algorithm in future studies. Thank you very much for your suggestion.

*Line 319:   It would be good to explicitly document how the aerosol and cloud radiative forcing is computed in Section 2.1 since it is part of the product output.*

A. The following description for computation of ARF and CRF have been added to section 3.

'Aerosol radiative forcing (ARF) and cloud radiative forcing (CRF) are calculated as the difference between the radiative fluxes with and without aerosols or clouds, respectively. Specifically, ARF is defined as the difference between the radiative flux calculated with all aerosol components included and the flux calculated without aerosols. Similarly, CRF is defined as the difference between the radiative flux with all cloud components included and the flux calculated in the absence of clouds. These calculations are performed for both the TOA and the SFC to assess the impact of aerosols and clouds on the Earth's energy budget.'

*Figure 1: What is the wavelength for the extinction shown in panels "c"?   Panel "e" is a bit hard to follow.   Could it be split into a panel for SW and a panel for LW?   For the current panel "e", the "obs" legend markers at the bottom of the plot are barely visible.   It would also be good to have panel "e" aligned along the x-axis with the panels above it.   Also, it is quite challenging to compare the markers for the computed and observed fluxes since they are fluctuating significantly, perhaps a line plot would be better.*

A. The wavelength in panel (c) is 532nm, and it has been added to the figure. We have divided Figure 1 (e) into two separate panels: one for SW (panel (e)) and one for LW (panel (f)). This division makes it easier to see the value fluctuations and markers. Thank you for your suggestion.

*Figure 8: It is difficult to see any structure to the cloud forcing on the plots. It would be helpful to consider modifying the plots so that some of the structure can be seen.*

A . Widening the color bar range would make it difficult to capture the subtle effects in the LW radiation. Since the primary goal of this study is to demonstrate the overall cooling effect in both SW and Net radiation at the TOA and SFC, we would like to retain the current color bar settings. Also, the spatial pattern of heating and cooling in ATM for LW is clearly visible within the current color bar range.

---

## Author Comment (AC2)

**Response to Referee #2**

Thank you very much for your time and effort taken to review our manuscript submitted to AMT. We really appreciate the reviewers' constructive comments that are very useful to greatly improve the manuscript. We have revised the manuscript based on your comments as explained below. Please see below for our point-by-point responses to your comments, where the original comments are shown in *italics* and our responses are shown in normal text just below your corresponding comments.

*Specific Comments:*

*Line 85: It is not clear how the target accuracy is set as 10 W m-2. Certainly, the SW biases shown in this study are higher than this target number. Is this target number based on existing products? If so, please include the relevant references.*

A. We have added a quote of (ESA, 2001) that demonstrates the scientific goals of the EarthCARE mission.

*Line 113: Do these two sets of satellite products (CloudSat/CALIPSO/MODIS) and (CPR/ATLID/MSI) provide consistent cloud and aerosol parameters? The algorithm developed in this study was tested using the A-train products. Therefore, it is important whether those two sets of products have comparable parameters.*

A. Yes, (CloudSat/CALIPSO/MODIS) and (CPR/ATLID/MSI) provide consistent cloud and aerosol parameters. The following text was added.

'CloudSat/CALIPSO/MODIS and CPR/ATLID/MSI will provide consistent cloud and aerosol parameters.'

*Line 123-124: It is not clear how the attenuated backscatter coefficient and depolarization ratio were used to derive the vertical profiles of three aerosol types. Please include the relevant references or description of it. In addition, what are specifically vertical profiles of aerosol types? Are these aerosol extinction profiles, single scattering albedo, and asymmetry parameters?*

A.  The fine-mode spherical particle (WS), coarse-mode spherical particle (SS), and non-spherical particle (DS) are classified by using the ratio of attenuated backscatter coefficient at 532 nm and 1064 nm and depolarization ratio of the CALIPSO measurements. The ratio of attenuated backscatter coefficient at 532 nm and 1064 nm depends on the aerosol particle size and the depolarization ratio depends on the aerosol particle shape. The size and optical properties of these three aerosol components are listed in Nishizawa et al. (2011).

We have added the following sentences to the text in the revised manuscript.

'The extinction coefficient of fine-mode spherical particle (WS), coarse-mode spherical particle (SS), and non-spherical particle (DS) are derived from the CALIPSO observation. The vertical profiles of extinction coefficient at 532 nm for WS, DS, and SS are used in the radiative transfer calculations. The particle size and optical properties of these three aerosol components are listed in Nishizawa et al. (2011).'

Nishizawa, T., Sugimoto, N., Matsui, I, Shimizu, A., and Okamoto, H.: Algorithms to retrieve optical properties of three component aerosols from two-wavelength backscatter and one-wavelength polarization lidar measurements considering nonsphericity of dust, J. Quant. Spectrosc. Radiat. Transfer., 112, 254-267, 2011.

*How was the MODIS COT used for constraining cloud radiative properties? Please provide detailed information.*

A. When there is a discrepancy between the COT derived from the vertical information of CloudSat/CALIPSO and the COT from MODIS, the vertical extinction coefficient is adjusted to align with the COT from MODIS. We have included this information as follows in the revised manuscript.

 'When there is a discrepancy between the COT derived from the vertical information of CloudSat/CALIPSO and the COT from MODIS, the vertical extinction coefficient is adjusted to align with the COT from MODIS.'

*Line 138: Is the GEOS-4,5 different from MERRA-1 or MERRA-2?*

A. The meteorological field variables from GEOS-5, which were stored in CCCM product (Kato et al. 2011), are used.

*Line 140: Please provide the information about the surface emissivity assumptions used for RT calculations. Was the skin temperature from GEOS?*

A. The surface emissivity varies with the type of ground surface (sea, land, or sea ice). Yes, the skin temperature from GEOS data was used. We have included this information in the revised manuscript.

'The meteorological field variables (pressure, temperature, and specific humidity, and skin temperature) from NASA's Goddard Earth Observing System (GEOS-5) Data Assimilation System (Bloom et al., 2005; Rienecker et al., 2008) are used in the radiative transfer calculations.'

*Line 143: For the area of 5 km, is it assumed as completely clear and cloudy? I think the homogenous assumption would be okay for most cases, but for partly cloudy cases, this homogeneous assumption can cause positive SW biases, as discussed in earlier 3D cloud studies.*

A. When verifying, the all-sky conditions include instances where the 5-km area is a mixture of clear and cloudy conditions, while the cloudy case extracts only instances where the entire 5-km area is covered by clouds. The TOA flux is compared to CERES with a footprint of 20 km, so in cloudy conditions all 20 km are covered by clouds. We have included this information in the revised text as follows.

'When verifying, the all-sky conditions include instances where the 20 km area was a mixture of clear and cloudy conditions, while the cloudy conditions extract only instances where the entire 20 km area was covered by clouds'

*Line 154: Which CERES product was used for the observed TOA fluxes?*

**A.** We used the CERES data that is included within the CCCM (CALIPSO-CloudSat-CERES-MODIS) product. We have added the specific product name, CER_ES8_Aqua-FM3_Edition3, in the revised manuscript to clarify the source of the CERES data used in our study.

*Line 154 or later in the result section: The surface radiation significantly varies by region as the authors noted. Therefore, it would be helpful where the ground sites are located.*

**A.** A location map of BSRN observation sites (please see the figure below) was created and added as Figure 6e in the revised manuscript.

[Figure]

*Line 155: The results in this study were compared with two versions (R04 and R05) of the 2B-FLXHR-Lidar product. Therefore, it would be necessary to provide a brief description of how these versions differ.*

**A.** In 2B-FLXHR-Lidar R05, the input values for clouds and aerosols have been updated to use the R05 versions of the CloudSat products. These updates include improvements in cloud coverage, cloud physical properties, including updated cloud phase information, and the use of

CALIPSO V4 products for aerosols, which update the global distribution of aerosol types and aerosol optical depth (AOD). These enhancements allow for more accurate flux calculations.

We have added these descriptions to the text in the revised manuscript.

*Line 157: Those four months were used for validations at TOA and surface? I was wondering why the sampling number is so small for the ground comparison.*

A. Those four months were used for TOA and surface validation. The number of surface samples is very small because the observations used for comparison are limited to within $\pm0.1$ degrees of the A-Train orbit for matchup of data, and also due to the limited target period.

*Eq. (3): I don't see any comparison of heating rate profiles, besides the example shown in Fig. 1d.*

A. Since the heating rate calculation is derived from the radiative flux, this study focuses on comparing the radiant flux. However, future studies will be needed to include the comparisons of heating rate to and to discuss any differences of our product from others.

*Line 183: I believe that the CERES CCCM product also provides flux at 20 km resolution. Have the authors compared the results with what this product provides?*

A. This study has not done the comparisons with the CCCM product. We would like to extend the comparisons with other products to include CCCM as well in our future studies.

*Line 216: Does it mean that each point in the scatter plots was from monthly 5-degree grided points for four months in 2007?*

A. Yes, that's correct. Each point in the scatter plots represents data from monthly 5 ° gridded points over four months in 2007. This statement has been added to the text as follows: "Each point in the scatter plots represents data from monthly 5 ° gridded points over four months in 2007."

*Line 425: Figure 3 is the same as "Figs 2a and 2d" but separated by cloud types. I guess that the 2B-FLXHR-Lidar fluxes are not included in Fig. 3. If so, the scatter plots shown in Fig. 3a–3e are subsets of Fig. 2a? Likewise, the scatter plots shown in Figs. 3f–j are subsets of Fig. 2d? Please clarify it in the figure caption. Why some outliers shown in Fig. 3g are not shown in Fig. 2d?*

A. Figure 3 is a subset of Figure 2a and 2d, showing only the cloudy cases. Since the cloud phase classification is based on MODIS observations, the comparison is limited to day-time cases only. Therefore, the outliers in Figure 3g disappear when night-time and clear-sky cases are included in the averaging process. We have revised the figure caption to incorporate these points.

*Line 244-246: If the consistent ice scattering model (i.e., Voronoi-type) was used for cloud retrievals and RT calculations, this would not be a problem. Please include more discussion about it.*

A.    The Voronoi-type model is consistent between the retrievals and the RT calculations, but there might be other issues occurring during the retrievals or radiative transfer calculations. Our argument here intends to mean that the COT retrieval for ice clouds might be a candidate source of error, particularly given that the Voronoi assumption is common between our RT simulation and the MODIS retrieval. This is something that will need to be further investigated in future work. To clarify the point above, we have revised the text of this part as follows: "The positive SW bias could have been caused by a possible overestimation of the ice cloud optical thickness obtained from MODIS, particularly given that the assumption of Voronoi-type ice particles is common among the radiative transfer simulation and the MODIS retrieval of ice cloud optical thickness."

*Line 249: Was the sensitivity study using NASA MAC06S0 performed using a consistent ice scattering model between cloud retrievals and RT calculations?*

A.  In NASA's sensitivity experiments, ice scattering is not consistent. However, COT tends to be lower in NASA's products compared to JAXA's products. This difference in COT is interpreted to contribute to the reduction of the positive bias.

*Line 253: "LW bias by providing more accurate cloud detection through improved measurement instrumentation" It is not clear what this statement specifically refers to. Please provide relevant references or expand the discussion.*

A. We appreciate the opportunity to clarify this statement. The reference to "providing more accurate cloud detection through improved measurement instrumentation" specifically refers to advancements in satellite-based remote sensing technologies, such as those utilized by EarthCARE, which offer enhanced cloud detection capabilities compared to earlier instruments. These advancements include higher resolution measurements and more sensitive detection of cloud properties, particularly those significant in the LW spectrum, which would contribute to reducing biases in LW radiative flux calculations. We have expanded the discussion to include relevant references and provide a more detailed explanation as follows: "Such advancements are expected particularly from technologies employed by the EarthCARE mission, which utilize improved instrumentation with higher spatial and spectral resolution, as well as enhanced sensitivity in detecting cloud properties, especially those significant in the LW spectrum. For example, EarthCARE's advanced radar and lidar systems allow for more precise cloud profiling, which leads to more accurate detection and characterization of cloud cover and thickness. This improved accuracy in cloud detection helps reduce biases in LW radiative flux calculations by ensuring that cloud-related inputs to radiative transfer models are more representative of actual atmospheric conditions."

*Line 299: As mentioned earlier, it would be helpful if the authors could provide the location of the BSRN sites on a map.*

A. A map of BSRN observation sites was created and added as Figure 8e in the revised manuscript.

*Line 299: What is "a minor bias"?*

A. By "a minor bias," we are referring to a small negative bias observed in the data.

---

## Author Comment (AC3)

**Response to Referee #3**

Thank you very much for your time and effort taken to review our manuscript submitted to AMT. We really appreciate the reviewers' constructive comments that are very useful to greatly improve the manuscript. We have revised the manuscript based on your comments as explained below. Please see below for our point-by-point responses to your comments, where the original comments are shown in *italics* and our responses are shown in normal text just below your corresponding comments.

*The manuscript describes the algorithm to compute top-of-atmosphere and surface radiative fluxes and radiative flux profiles in the atmosphere using measurements from EarthCERE instruments, CPR, ATLID, MSI, and BBR. The algorithm was developed by the EarthCARE Japanese group. TOA flux is used to evaluate retrieved properties by comparing fluxes with fluxes derived from BBR. The EarthCERE's goal is to achieve the difference less than 10 Wm-2. The authors test the algorithm using A-train data. When instantaneous fluxes are averaged in 5-degree grids and over a month, the bias and RMS difference compared with CERES derived fluxes are, respectively, 24 Wm-2 and 36 Wm-2 for shortwave and -11 Wm-2 and 14 Wm-2 for longwave. The purpose of the manuscript is to describe the algorithm and evaluation of the algorithm. While the manuscript meet this goal, it does not provide new information other than these purposes. Using 1D radiative transfer and comparing with observed TOA fluxes with A-train data are not new. New science results are missing. In addition, given similar products are available from the European team, the manuscript needs to highlight unique aspects of the Japanese flux products, distinguishing from European products.*

A. Thank you very much for your comments. The 1D radiation calculation is used to develop the present algorithm to meet the requirements for data delivery latency in JAXA's standard product generation. For the research product, which is another data category in JAXA without such latency requirements differently from the standard product, we plan to provide the outcomes of 3D radiation calculations although it is beyond the scope of this paper. Our detailed responses to your comments are provided below including these points.

*Major comments.*

*The authors describe radiation budget, especially downward longwave radiation at the surface in the introduction section. However, given what the authors describe in this manuscript, I do*

*not see how the algorithm and data products described in this manuscript will contribute to improving global surface radiation budget and downward longwave, in particular, from the level where we are with A-train data. If Japanese flux products are to improve surface radiation, please describe how to improve in the manuscript.*

A. This paper primarily demonstrates the preparatory stage for the EarthCARE product, and once EarthCARE data becomes available, the cloud data will be replaced with those from EarthCARE. With the enhanced capabilities of CPR and ATLID, which will better capture low-level clouds, we expect to see improved contributions to downward longwave radiation as well. The following sentence has been added to the introduction.

'With the enhanced performance of EarthCARE's CPR and ATLID, which will better capture low-level clouds, we expect to see improved contributions to downward longwave radiation as well.'

*Similarly, the authors mention that aerosol and cloud vertical profiles affect vertical profile of radiative fluxes. The number of aerosol type is increased from three to four in the algorithm. This is still less that the number of aerosol types used in CALIPSO algorithms (see for example Omar et al. 2009; Burton et al. 2012, 2013) and flux computations. Please provide thoughts of how to improve our knowledge of vertical flux profiles with the flux products described in this manuscript.*

A. As aerosol species, fine-mode particle (WS), fine-mode and light-absorbing particle (LA), coarse-mode particle (SS), and coarse-mode and light-absorbing particle (DS) are assumed in the JAXA EarthCARE lidar retrievals. These four aerosol components are similar to aerosol species of the chemical transport model. The definitions of aerosol type are different between JAXA and NASA products. For example, smoke and polluted continental of NASA product (Omar et al. 2009) are consist of WS and LA. In addition, polluted dust of NASA product is the mixture of smoke and dust, which are consist of WS, LA, and DS. In this study, the light-absorption of aerosols emitted from biomass burning and air pollution may be underestimated, because of the lack of LA. The estimation of aerosol radiative effect will be improved in the JAXA EarthCARE product by including LA. **The following sentence has been added to the section 1.**

'By adding LA in the EarthCARE product, the estimation of light-absorption for biomass burning and air pollution, which include LA, will be improved and aerosol radiative effect is expected to be more accurately evaluated during the EarthCARE mission.'

*The introduction provides some background of surface radiation budget. EarthCARE data are, however, likely to contribute improving our knowledge of vertical flux profiles than improving global radiation budget.*

A. Although we believe that improving the vertical cloud coverage will enhance global radiation calculations, improvements in cloud physical quantities such as cloud water content and ice water content, as well as better aerosol characterization enabled by ATLID, will also lead to better estimates of vertical profiles of radiative fluxes, as the reviewer pointed out. We have added these points in the revised manuscript as follows: "Such enhanced information of Earth's energy budget will also be facilitated by improved knowledge of vertical profiles of radiative fluxes expected from the detailed cloud profiling capability combined with cloud dynamics information."

*The approach described in the manuscript has been used with A-train data for at least 10 years. Could you describe the uniqueness of the data products? What do they offer scientifically that is not available from European products and A-train products (e.g. FlxHR or CCCM)? Unless the authors describe clearly here, users are not motivated to use the Japanese products unless they are involved in the project.*

A. Thank you very much for raising these important points. For the past A-Train data, FLXHR and CCCM data can be utilized, but for EarthCARE, to our knowledge, there has been no announcement regarding the provision of radiative flux data beyond the Japanese and European data. Also, given our another development of 3D radiative transfer (RT) code, called MCstar, and its application to some cases of cloudy scene as described in Okata et al. (2017), it would be possible in the future to seamlessly compare 1D and 3D radiative calculations based on the common assumptions and settings of particulate and gaseous optical properties that are used in our 1D RT code (MstrnX used in this study) and 3D RT code (MCstar). This would allow for error quantifications of 1D RT against 3D RT and possible introduction of several methods for approximating 3D RT effects in the framework of 1D RT computation, as also described in Okata et al. (2017). We plan to incorporate these improvements into the standard algorithm with

1D RT described in the present paper, as well as development of the research product with 3D RT, so as to add values to our Japanese radiation products. Additionally, we also believe that comparing these Japanese data products based on the 1D and 3D RT computations with those from European side will lead to improvements in both datasets. These points are added in the revised manuscript at the beginning of Section 2.

Okata, M., T. Nakajima, K. Suzuki, T. Inoue, T. Y. Nakajima, and H. Okamoto, 2017: A study on radiative transfer effects in 3-D cloudy atmosphere using satellite data. J. Geophys. Res. Atmos., 122, 443−468, doi:10.1002/2016JD025441.

*It is not critical but given the bias at TOA, how the Japanese team is going to achieve the goal of EarthCARE of 10 Wm-2? In addition, this manuscript is revealing that the Japanese flux algorithm is more primitive compared to European flux algorithms. I think that it is useful for the international community having independent flux results from the Japanese and European teams. From this point, it is useful if the authors provide their thoughts on how the international community will benefit having the Japanese flux products in addition to Europeans.*

A. As mentioned in the previous response, we believe that comparing the Japanese and European products will contribute to the improvement of both, benefiting from having independent radiative flux algorithms from the Japanese and European teams, as the reviewer pointed out. By providing the algorithm development team for cloud physical properties with findings from this paper, such as the flux error characteristics of ice clouds, we expect to contribute to improvements in the cloud properties retrievals toward an achievement of the 10 W/m² accuracy. Furthermore, the European team has focused on analyzing specific cases, without conducting long-term analyses, which suggests that operationalizing 3D radiative transfer calculations may not be feasible for them. In this regard, we also plan to develop a flux algorithm based on 3D radiative transfer calculations (Okata et al., 2017) as part of our research products. The improvement of the 1D standard product is also planned through comparison with 3D radiation calculations in future studies, as mentioned in our previous response. The following sentence has been added to the text to reflect the argument above.

"it is worth noting that our previous study developed a three-dimensional (3D) radiative transfer code and applied it to some cases of cloudy scenes as described in Okata et al. (2017). It would then be possible in future studies to seamlessly compare the 1D and 3D radiative calculations based on the common assumptions and settings of particulate and gaseous optical properties that

are used in our 1D and 3D radiative transfer codes. This would allow for error quantifications of 1D against 3D radiative transfers and possible introduction of several methods for approximating 3D effects in the framework of 1D radiative transfer computation, as also described in Okata et al. (2017). In future studies, we plan to incorporate these improvements into the standard algorithm with 1D radiative transfer described in the present paper, as well as to develop a radiative flux algorithm based on 3D radiative transfer calculations (Okata et al. 2017) as part of the research product, so as to add values to our Japanese radiation products."

**Minor comments**

*Line 134: Could you explain what the Voronoi particles are?*

A. The Voronoi particles are particles that do not have regular spherical shapes, but rather particles with irregular polyhedral shapes. Voronoi particles are commonly used to model the scattering properties of ice crystals in clouds. For more details, please refer to the work of Ishimoto et al., (2010) which is cited in the manuscript. The following sentence has been added to the text in the revised manuscript to describe the Voronoi particles.

'The Voronoi particles are particles that do not have regular spherical shapes, but rather particles with irregular polyhedral shapes.'

*Line 247: Could you justify reducing the optical thickness by 30%?*

A. This is based on the bias estimate of the COT retrieval for ice clouds. Nakajima et al. (2019) show a COT bias of 2.4 for ice clouds relative to MODIS products, so a 30% reduction for ice clouds with small COT is considered reasonable. The following sentence has been added to the text.

'Nakajima et al. (2019), who described the cloud property retrievals from shortwave reflectance, showed a COT bias of about 2.4 for ice clouds relative to MODIS products, so that a 30% reduction of COT for ice clouds with small COT can be considered reasonable.'

*Line 253: If the authors claim EarthCARE instruments detect more clouds, then computed OLR is even lower, which increases the bias. Please explain why EarthCARE is expected to reduce the LW bias.*

A. EarthCARE's ATLID and CPR will provide more accurate vertical profiles of cloud properties, such as cloud phase and cloud-top height, which are crucial factors in determining LW radiation. These enhancement in accuracy of cloud microphysical properties, when used as input data for radiative transfer computation, is anticipated to lead to better estimates of longwave radiation.

*Line 330: Could you elaborate why aerosol radiative forcing is important in the upper atmosphere? Also, does the Japanese team retrieve aerosol properties everywhere all the time? What do you use when retrieved aerosol properties are not available (e.g. below clouds)?*

A. We intended to mean simply "atmosphere" contrasted against "surface", not specifically meaning the "upper atmosphere". Therefore, we have deleted the "upper" from the sentence in the revised manuscript. However, aerosols above clouds are important because they can induce multiple scattering between clouds and aerosols, thereby altering radiative effects. When aerosols are present above clouds, this interaction can significantly impact the overall radiative forcing of aerosols. Including such an "above-cloud aerosol" case, the vertical stratification of aerosols and clouds is a key factor in determining aerosol radiative forcing as Oikawa et al. (2013, 2018) have shown. The Japanese team's approach primarily uses aerosol data from ATLID, which allows for more detailed calculations of these interactions. However, when thick clouds are present, aerosol data cannot be retrieved, as the reviewer pointed out. For future research products, we are considering incorporating aerosol reanalysis products as input data, which would allow us to include aerosols below the cloud layer as well.

Oikawa, E., Nakajima, T., Inoue, T., and Winker, D.: A study of the shortwave direct aerosol forcing using ESSP/CALIPSO observation and GCM simulation, J. Geophys. Res. Atmos., 118, 3687–3708. https://doi.org/10.1002/jgrd.50227, 2013.

Oikawa, E., Nakajima, T., Inoue, T., and Winker, D.: "An evaluation of the shortwave direct aerosol radiative forcing using CALIOP and MODIS observations", J. Geophys. Res. Atmos., 123, 1211–1233, https://doi.org/10.1002/2017JD027247, 2018.

**References**

Burton, S. P., and coauthors, (2012). Aerosol classification using airborne high spectral resolution lidar measurements – methodology and examples, Atms. Meas. Tech., 5, 73-98.

Burton, S. P., Ferrare, R. A., Vaughan, M. A., Omar, A. H.,  Rogers, R. R., Hostetler, C. A., and Hair, J. W., (2013). Aerosol classification from airborne HSRL and comparison with the CALIPSO vertical feature mask, Atmos. Meas. Tech., 6, 1397-1421.

Omar, A., Winker, D., Kittaka, C., Vaughan, M., Liu, Z., Hu, Y., Trepte, C., Rogers, R., Ferrare, R., Kuehn, R., Hostetler, C., (2009). The CALIPSO Automated Aerosol Classification and Lidar Ratio Selection Algorithm, J. Atmos. Oceanic Technol., 26, 1994–2014.

---

## Author Comment (AC4)

**Response to Referee #4**

Thank you very much for your time and effort taken to review our manuscript submitted to AMT. We really appreciate the reviewers' constructive comments that are very useful to greatly improve the manuscript. We have revised the manuscript based on your comments as explained below. Please see below for our point-by-point responses to your comments, where the original comments are shown in *italics* and our responses are shown in normal text just below your corresponding comments.

*This manuscript describes the theoretical foundations of the Japanese radiative flux and heating rates product for EarthCARE. The algorithm derives vertical profiles of longwave (LW) and shortwave (SW) radiative fluxes and heating rates at 34 atmospheric levels by applying a radiative transfer model to aerosol and cloud profiles retrieved from the EarthCARE cloud profiling radar, lidar, and multi-spectral imager. The primary focus of this study is to document the anticipated accuracy of the product by applying the algorithm to existing observations collected by the A-Train. The subject is appropriate for Atmospheric Measurement Techniques €i0and the uncertainty analysis is quite thorough considering the algorithm has yet to be implemented for EarthCARE. My primary concerns center on the organization of the findings. In particular, the abrupt transition from the algorithm description to validation could be softened by including the preliminary results prior to discussing the comparisons. In addition, there are several opportunities to reference related literature that should be considered. Since I do not anticipate those modifications requiring substantial rewriting, I recommend the paper be published in AMT after the following minor revisions to address these concerns.*

A. We sincerely thank the reviewer for the thoughtful comments and valuable suggestions. We appreciate the recognition of the manuscript's contributions to the field and have carefully considered the feedback provided. Our responses to each of the reviewer's comments are detailed below.

*Specific Comments:*

1. *The most significant issue with the paper in its current form is the organization of results. This transition from algorithm description immediately into comparisons with CERES is quite abrupt. It would be interesting to see some examples of the algorithm before discussing its evaluation. I think the example in Figure 1 could be used to simply illustrate the methods described in Section 2 (omitting the CERES comparisons in panel (e) which are hard to see anyway). That could be followed the spatial distributions of aerosol and*

*cloud radiative effects in Figures 6 and 7 to provide context for what the algorithm does before assessing the accuracy of these results.*

A. The entire text was reorganized following the reviewer's suggestion, with the demonstration of input and output referring to Figure 1 moved to Section 2 and the section on cloud and aerosol radiative forcing moved to Section 3. Figure 1(e) is drawn separately for SW and LW to make the plot easier to see. Some additional adjustments of texts have also been done for a smooth transition from description of methodology, through demonstration of aerosol and cloud radiative forcing, to evaluation against CERES and BSRN. We believe that the presentation became much smoother than the previous version. Thank you very much for your suggestion.

*1.      Line 43: The acronym for CERES is missing some words "Clouds and the Earth's Radiant Energy System"*

A. We have added the phrase 'Clouds and the' as per your suggestion.

*2.      Line 49: Since this is not the first paper to estimate fluxes using radiative transfer modeling with atmospheric inputs, I suggest referencing some of the pioneering papers on this topic (e.g. Rossow and Lacis, 1990; Rossow and Zhang, 1995; Zhang et al, 1995; Whitlock et al, 1995).*

A. The references have been added as suggested. Thank you very much for suggesting these literatures.

*3.      Line 69: It may also be worth adding that these measurements will provide important continuity for the data record that began with the A-Train in 2006 (L'Ecuyer and Jiang, 2010).*

A. In response to your comment, we have added the following sentence to highlight the importance of these measurements in terms of the data record continuity that began with the A-Train in 2006, as discussed by L'Ecuyer and Jiang (2010).

'These measurements will also provide important continuity for the long-term data record that began with the A-Train in 2006 (L'Ecuyer and Jiang, 2010), ensuring that trends and patterns in atmospheric observations are consistently maintained.'

*4.      Line 90: While it is likely beyond the scope of this particular study, there could be value in digging deeper into comparisons with FLXHR-lidar and CCCM to trace the source*

*of discrepancies in all three algorithms. Since the algorithm has already been applied to CloudSat/CALIPSO/MODIS observations, it could immediately be compared to FLXHR-lidar and CCCM in a manner like that of Ham et al. (2014). The results would be very interesting for understanding all three algorithms.*

A. We agree that a deeper comparison with FLXHR-lidar and CCCM would be valuable for tracing the source of discrepancies between the three algorithms. However, as this is beyond the scope of the current study, we have not included this analysis in the present paper. We do recognize the importance of this comparison and intend to pursue it as part of our future work. This will allow for a more comprehensive understanding of the differences and similarities among the algorithms, as highlighted by Ham et al. (2017). The following sentence has been added to the text.

'Ham et al. (2017) compared CCCM with 2B-FLXHR-Lidar, showing regional differences in radiative fluxes due to differences in cloud characteristics within the products, and we believe that more detailed comparisons between products, including our product, would be beneficial and needed to further improve the products as future work.'

*5. Line 113: I think 'were utilized' should be 'will be utilized' since EarthCARE data were not actually used in this paper.*

A. The text has been revised following the reviewer's comment. Thank you for correcting our English.

*6. Line 157: Do you mean 'daytime' instead of 'diurnal'?*

A. Corrected to 'daytime'. Thank you.

*7. Line 180 (and again on Line 311): The spatial resolution of CloudSat is 1.4 km (across track) by 1.8 km (along track).*

A. Spatial resolution was corrected to '1.8 km'.

*8. Line 229 - 231: There is precedence for separating results according to cloud phase in this way. Perhaps cite Matus et al. (2017) here.*

A. Matus and L'Ecuyer (2017) was added to cite here.

*9. Line 297: The preceding discussion does not provide adequate context for the value of these estimates. The ability of spaceborne active sensors to constrain surface fluxes and atmospheric flux divergence represents one of the most important contributions they*

*have made to climate science. This is discussed in detail in papers like Haynes et al. (2010), L'Ecuyer et al. (2019), and Hang et al. (2019), for example. If this is better articulated in the introduction, the point here could be that without quantifying the uncertainties, it is hard to know how trustworthy this information is.*

A. Thank you very much for this suggestion to better motivate our study. The following sentences were added to the introduction and Section 5, respectively, in the revised manuscript.

Introduction

'In addition, spaceborne active sensors have made significant contributions to climate science by providing more precise constraints on atmospheric and surface radiative fluxes compared to passive sensors. These active sensors play a crucial role in improving climate models by offering more accurate measurements of radiative fluxes and heating rates partitioned into atmosphere and surface (Haynes et al., 2013; L'Ecuyer et al., 2019; Hang et al., 2019). However, without quantifying the uncertainties, it is difficult to fully evaluate the reliability of these estimates of radiation based on active sensors. Therefore, one of the objectives of this study is to assess these uncertainties through comparisons with other products and ground-based observations, aiming to validate the accuracy and reliability of the radiative flux based on the active sensor.'

Section 5

'These findings highlight the importance of spaceborne active sensors in constraining surface and atmospheric fluxes, which are essential for accurate climate modeling. However, without quantifying the uncertainties associated with these estimates, it is challenging to fully trust the information they provide. Therefore, the quantification of uncertainties is crucial to assess the reliability of the derived fluxes and their implications for climate science.'

*10.      Line 328: It would also be good to compare against other recent studies that produce similar estimates (Matus et al, 2019 is one example but there are others, including some by Winker et al.)*

A. We have added the following text that includes a comparison to Matus et al. (2019). Thank you for your suggestion.

'Our study's results align with those of Matus et al. (2019), who reported a global mean aerosol direct radiative effect (DRE) of $-2.40$ W/m², primarily driven by sulfate aerosols with significant uncertainty due to aerosol type classification and optical depth retrievals. Similarly, our findings emphasize the critical role of accurate aerosol classification in determining the radiative forcing. Matus et al. (2019) also highlighted that anthropogenic aerosols contribute significantly to the

global radiative effect, estimating an anthropogenic direct radiative forcing (DRF) of −0.50 W/m². Our study corroborates these findings, further illustrating the substantial impact of anthropogenic aerosols on the Earth's energy budget. Both studies underscore the value of leveraging satellite-based observations to capture aerosol radiative effects, particularly in regions where ground-based measurements are sparse.'

11.     *Line 337: Similarly, some qualitative comparisons against prior work are warranted here as well (there are lots of options but Matus et al, 2017; L'Ecuyer et al, 2019 and Hang et al, 2019 all utilize similar observations to extract the effects clouds at TOA, SFC, and in the ATM).*

A. The following sentence was added to Section 3.

'Our findings on cloud radiative forcing are consistent with those reported in previous studies, including Matus and L'Ecuyer (2017), L'Ecuyer et al. (2019), and Hang et al. (2019). These studies similarly identified significant impacts of clouds on radiative forcing at the top of the atmosphere, surface, and within the atmosphere, supporting the robustness of our results.'

12.     *Line 363: This isn't an accurate statement. The analysis quantifies how the accuracy of radiative flux calculations varies with spatial and temporal averaging scale.*

A. The text has been corrected to be accurate as follows. Thank you very much.

'we quantified how the accuracy of radiative flux calculations varies with different spatial and temporal averaging scales.'

13.     *Figure 1: The transition from yellow to light blue in the upper atmosphere in Figure (d) is likely an artifact of the color bar. It might be good to have a small band of white from -0.05 to 0.05 to represent areas of 0 heating.*

A. Thank you for the suggestion. Here, as we aim to distinguish between cooling and heating, we prefer not to introduce a white band around the 0 value, and would like to retain the current color bar that effectively separates the heating and cooling.

14.     *Figure 3 caption: Technically this figure is only the same as Figure 2 panels (a) and (d).*
A. Added '(a) and (d)' to the caption.

15.     *Figure 5 caption: Again, this figure is only the same as Figure 4 panels (a) and (d).*
A. Added '(a) and (d)' to the caption.

*16.     There are also several minor grammatical errors throughout the paper. A few representative examples follow, but I suggest taking a careful read through the paper for other similar issues:a.Line 38: 'circulation' should be 'circulations'b.Line 44: 'radiometer' should be 'radiometers'c.Line 199: 'value' should be 'values'd.Line 200: 'of the aerosols' should be 'of aerosols'*

A. The grammatical errors have been corrected. Thank you.

References

1. Ham, S.-H., S. Kato, F. G. Rose, D. Winker, T. L'Ecuyer, G. G. Mace, D. Painemal, S. Sun-Mack, Y. Chen, and W. F. Miller, 2017: Cloud occurrences and cloud radiative effects (CREs) from CCCM and CloudSat radar-lidar products, J. Geophys. Res. 122, 8852-8884.

2. Hang, Y., T. S. L'Ecuyer, D. Henderson, A. V. Matuss, and Z. Wang, 2019: Reassessing the role of cloud type in Earth's radiation budget after a decade of active spaceborne observations. Part II: Atmospheric heating, J. Climate 32, 6219-6236.

3. Haynes, J. M., T. H. Vonder Haar, T. L'Ecuyer, and D. Henderson, 2013. Radiative heating characteristics of Earth's cloudy atmosphere from vertically resolved active sensors, Geophys. Res. Letters 40, doi:10.1002/grl.50145.

4. L'Ecuyer, T. S. and J. Jiang, 2010: Touring the Atmosphere Aboard the A-Train, Physics Today 63 (7), 36-41.

5. T. S. L'Ecuyer, Hang, Y., A. V. Matus, and Z. Wang, 2019: Reassessing the role of cloud type in Earth's radiation budget after a decade of active spaceborne observations. Part I: Top of atmosphere and surface, J. Climate 32, 6197-6217.

6. Matus, A. and T. S. L'Ecuyer, 2017: The role of cloud phase in Earth's radiation budget, J. Geophys. Res. 122, doi:10.1002/2016JD025951.

7. Matus, A. V., T. S. L'Ecuyer, D. S. Henderson, and T. Takemura, 2019: New global estimates of aerosol direct radiative effects, kernels, and forcing, from active satellite observations, Geophys. Res. Letters 46, 8338-8346.

8. Rossow, W. B., and A. A. Lacis, 1990: Global, seasonal cloud variations from satellite radiance measurements. Part II: Cloud properties and radiative effects, J. Clim., 3, 1204–1253.

9. Rossow, W. B., and Y.-C. Zhang, 1995: Calculation of surface and top of the atmosphere radiative fluxes from physical quantities based on ISCCP data sets: 1. Validation and first results, J. Geophys. Res., 100, 1167–1197.

10. Whitlock, C. H., and Coauthors, 1995: First Global WCRP Shortwave Surface Radiation Budget Dataset. Bull. Amer. Meteor. Soc., 76, 905–922, https://doi.org/10.1175/1520-0477(1995)076<0905:FGWSSR>2.0.CO;2.

11. Zhang, Y.-C., W. B. Rossow, and A. A. Lacis, 1995: Calculation of surface and top of the atmosphere radiative fluxes from physical quantities based on ISCCP data sets: 1. Method and sensitivity to input data uncertainties, J. Geophys. Res., 100, 1149–1165.